# Direct glia-to-neuron transdifferentiation gives rise to a pair of male-specific neurons that ensure nimble male mating

Laura Molina-García[1†], Carla Lloret-Fernández[1†], Steven J Cook[2‡], Byunghyuk Kim[3§], Rachel C Bonnington[1], Michele Sammut[1], Jack M O'Shea[1#], Sophie PR Gilbert[1], David J Elliott[1], David H Hall[3], Scott W Emmons[2,3], Arantza Barrios[1*], Richard J Poole[1*]

[1]Department of Cell and Developmental Biology, University College London, London, United Kingdom; [2]Dominick P. Purpura Department of Neuroscience, Albert Einstein College of Medicine, New York, United States; [3]Department of Genetics, Albert Einstein College of Medicine, New York, United States

*For correspondence:
a.barrios@ucl.ac.uk (AB);
r.poole@ucl.ac.uk (RJP)

[†]These authors contributed equally to this work

Present address: [‡]Department of Biological Sciences, Columbia University, New York, United States; [§]Department of Life Science, Dongguk University-Seoul, Goyang, Republic of Korea; [#]Centre for Discovery Brain Sciences, Edinburgh University, Edinburgh, United Kingdom

**Competing interests:** The authors declare that no competing interests exist.

**Abstract** Sexually dimorphic behaviours require underlying differences in the nervous system between males and females. The extent to which nervous systems are sexually dimorphic and the cellular and molecular mechanisms that regulate these differences are only beginning to be understood. We reveal here a novel mechanism by which male-specific neurons are generated in *Caenorhabditis elegans* through the direct transdifferentiation of sex-shared glial cells. This glia-to-neuron cell fate switch occurs during male sexual maturation under the cell-autonomous control of the sex-determination pathway. We show that the neurons generated are cholinergic, peptidergic, and ciliated putative proprioceptors which integrate into male-specific circuits for copulation. These neurons ensure coordinated backward movement along the mate's body during mating. One step of the mating sequence regulated by these neurons is an alternative readjustment movement performed when intromission becomes difficult to achieve. Our findings reveal programmed transdifferentiation as a developmental mechanism underlying flexibility in innate behaviour.

## Introduction

The coordinated execution of innate, stereotyped sexual behaviours, such as courtship and mating, requires sexually dimorphic sensory-motor circuits that are genetically specified during development (reviewed in *Auer and Benton, 2016*; *Barr et al., 2018*; *Yang and Shah, 2014*). Studies in the nematode *Caenorhabditis elegans*, in which the development and function of neural circuits can be interrogated with single cell resolution, have revealed two general developmental mechanisms underlying sexual dimorphism in the nervous system. The first involves the acquisition of sexually dimorphic features in sex-shared neurons during sexual maturation, which include changes in terminal gene expression, such as odorant receptors, neurotransmitters and synaptic regulators (*Hilbert and Kim, 2017*; *Jarrell et al., 2012*; *Oren-Suissa et al., 2016*; *Ryan et al., 2014*; *Serrano-Saiz et al., 2017a*; *Serrano-Saiz et al., 2017b*; *Pereira et al., 2019*; *Weinberg et al., 2018*). The second mechanism involves the generation of sex-specific neurons (*Sammut et al., 2015*; *Sulston and Horvitz, 1977*; *Sulston et al., 1980*). Sex-specific neurons are primarily involved in controlling distinct aspects of reproductive behaviours, such as egg-laying in the hermaphrodite and mating in the male (reviewed in *Emmons, 2018*). Generation of sex-specific neurons requires sex-specific cell death (*Sulston et al., 1983*) or neurogenesis events resulting from sex differences in the cell division patterns and neurodevelopmental programmes of post-embryonic cell lineages

(reviewed in *Barr et al., 2018*). Here we identify a novel way in which sex-specific neurons are generated in the nervous system: through a direct glia-to-neuron transdifferentiation of sex-shared cells.

In one of his seminal papers, John Sulston described a sexual dimorphism in the phasmid sensilla of adult animals (*Sulston et al., 1980*). The phasmid sensillum is one of the seven classes of sense organs that are common to both sexes in *C. elegans*. These sense organs are organised in sensilla which are concentrated in the head and the tail (*Bird and Bird, 1991*; *Ward et al., 1975*; *White et al., 1986*; *Doroquez et al., 2014*). Each sensillum is composed of the dendrites of one or more sensory neurons enveloped by a channel, usually composed of a single sheath glial cell and a single socket-glial cell. These sensilla can be viewed as part of an epithelium, continuous with the skin, and are shaped by mechanisms shared with other epithelia (*Low et al., 2019*). Socket-glial cells are highly polarised and adhere to the hypodermis at the distal end of their process where they form a small, ring-like hollow pore in the cuticle through which the sensory dendrites can access the outside world. The bilateral phasmid sensilla (*Figure 1*), situated in the tail, are unusual in that they are each composed of two socket-glial cells (PHso1 and PHso2). John Sulston observed that in juvenile animals (L2 stage) of both sexes, PHso1 forms the primary pore (*Sulston et al., 1980*; *Figure 1A*). At adulthood, the hermaphrodite retains a similar structure (*Figure 1B*). In males, however, it is PHso2 that forms the main pore and PHso1 was described as having retracted from the hypodermis and to protrude into the phasmid sheath (*Figure 1C*). It was also described to contain basal bodies, a structural component of cilia, in the region enveloped by the phasmid sheath. As sensory neurons are the only ciliated cells in *C. elegans* (*Perkins et al., 1986*), this is suggestive of neuronal fate, yet Sulston observed no other neuronal characteristics (*Sulston et al., 1980*). We previously showed that in the amphid sensillum (a similar organ located in the head) the amphid socket-glial cell (AMso) acts as a male-specific neural progenitor during sexual maturation, dividing to self-renew and generate the MCM neurons (*Sammut et al., 2015*). We therefore sought to investigate the PHso1 cells in more detail.

We find that during sexual maturation (L4 stage), the pair of sex-shared PHso1 glial cells acquire sexually dimorphic function by undergoing a direct (without cell division) glia-to-neuron transdifferentiation that results in the generation of male-specific neurons, which we term the phasmid D neurons (PHDs). This cell-fate plasticity is regulated by the sex-determination pathway, likely cell-intrinsically. However, we find the cell-fate plasticity does not depend on certain molecular

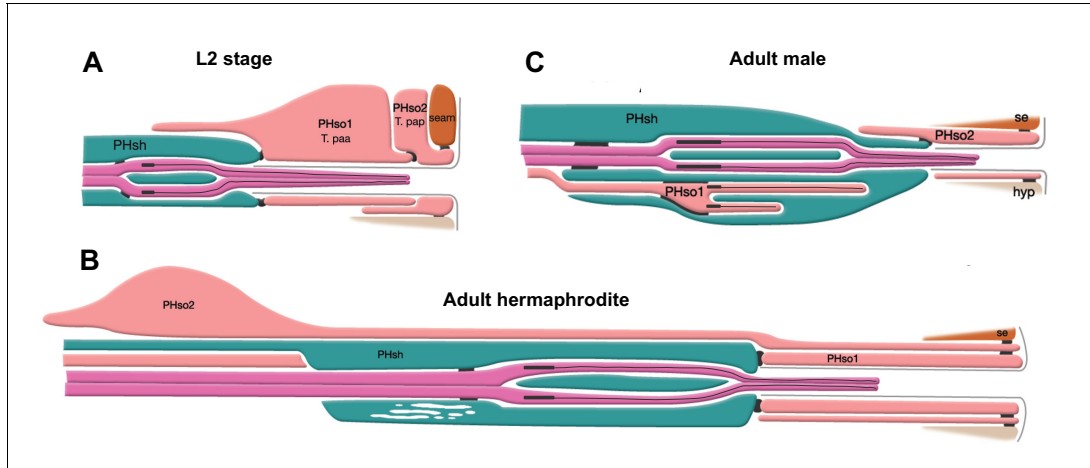

**Figure 1.** The phasmid sensillum. Diagram of the phasmid sensillum in either sex at the L2 larval stage (**A**), in adult hermaphrodites (**B**) and in adult males (**C**). The socket-glial cells (PHso1 and PHso2) are coloured in light pink; the sheath glial cells (PHsh) in green; and the ciliated dendrites of the phasmid sensory neurons, in dark pink. The adherens junctions are depicted as black lines between cells. Axonemes and cilia are marked as black bars and black lines inside the dendrite tips. Each phasmid opens to the exterior on the extreme right (posterior), where grey lines mark the cuticle borders of the phasmid pore and fan. Hypodermis (hyp), seam (se). Diagram has been modified from and is used with permission from http://www.wormatlas.org.

mechanisms known to regulate the only other well-described direct transdifferentiation in *C. elegans* (*Kagias et al., 2012*). This bilateral pair of previously unnoticed neurons are putative proprioceptors that regulate male locomotion during specific steps of mating. One of these steps is a novel read-justment movement performed when intromission becomes difficult to achieve. Our results reveal sex-specific direct transdifferentiation as a novel mechanism for generating sex-specific neurons and also show the importance of proprioceptive feedback during the complex steps of mating for successful reproduction.

## Results

### The sex-shared PHso1 cells undergo glia-to-neuron transdifferentiation in males

Using a *lin-48/OVO1* reporter transgene to identify and visualise the PHso1 cell bodies both before and during sexual maturation (*Johnson et al., 2001*; *Wildwater et al., 2011*), we observed no distinguishable differences between the hermaphrodite and male PHso1 cells at the L3 stage (*Figure 2A*). The PHso1 cells display a polarised morphology and a visible socket structure in both sexes. In hermaphrodites, this morphology is maintained during the transition to adulthood and PHso1 cells elongate as the animal grows. In males, by contrast, the PHso1 cells display morphological changes that can be observed during the L4 stage, the last larval stage before adulthood (*Figure 2A*). We find that during this stage PHso1 retracts its socket process and extends a short dendrite-like posterior projection into the PHsh cell and a long axon-like anterior process projecting towards the pre-anal ganglion region. For this reason and those described below, we use the name PHD to refer to the PHso1 cells after these morphological changes take place. A time-lapse of an individual male reveals that socket retraction is initiated at early L4, when the male gonad has almost reached the tail, and that the nascent axon is first observed at mid-L4, when the vas deferens has joined with the cloaca (*Figure 2B*). The axon-like outgrowth is complete by the end of tail-tip retraction, when ray precursor cells start to fuse into the tail seam syncytium (*Figure 2—figure supplement 1*). These observations corroborate those of John Sulston and identify the stage of sexual maturation as the time during which the PHso1 cells undergo radical remodelling in males. This remodelling involves quite a remarkable change in morphology from a socket-glial morphology to a neuron-like morphology.

To determine whether the PHso1 cells also acquire neuronal characteristics at the level of gene expression, we analysed and compared the expression of glial and neuronal markers in the PHso1 and PHso2 cells of males and hermaphrodites. In adult hermaphrodites, we find that both cells express the panglial microRNA *mir-228* (*Wallace et al., 2016*) and the AMso and PHso glial subtype marker *grl-2* (a Hedgehog-like and Ground-like gene protein: *Hao et al., 2006*) at similar levels (*Figure 3A*). In males, by contrast, *mir-228* and *grl-2* expression is noticeably dimmer in PHso1 than in PHso2 by the mid-to-late L4 stage and completely absent by day 2 of adulthood (*Figure 3A, B and C*). Importantly, expression of these reporters appears equal in brightness in both cells at the late L3 stage and only becomes noticeably dimmer in PHso1 following the morphological changes, such as the retraction of the socket, that occur during the L4 stage (*Figure 3B and C*, *Figure 3—figure supplement 1*). In contrast, in hermaphrodites, no morphological changes of PHso1 are observed and the reporters appear equal in brightness through to adulthood (*Figure 3A*, *Figure 3—figure supplement 1*). After the posterior hypodermal cells begin to retract leaving the characteristic fluid-filled extracellular space at the tail, PHso1 begins to express the pan-neuronal marker *rab-3* (a synaptic vesicle associated Ras GTPase: *Stefanakis et al., 2015*) in males but not in hermaphrodites, and this expression persists throughout adulthood (*Figure 3A, B and C*). In addition to *rab-3* expression we also observe sequential expression of other neuronal markers such as: the vesicle acetylcholine transporter *unc-17* (*Alfonso et al., 1993*; *Pereira et al., 2015*; *Figure 3D*); the phogrin orthologue for dense-core vesicle secretion *ida-1* (*Zahn et al., 2001*; *Figure 3E*); the intraflagellar transport component required for proper sensory cilium structure *osm-6* (*Collet et al., 1998*; *Figure 3F*); and the IG domain containing protein *oig-8* (*Howell and Hobert, 2017*; Figure 6A). None of these neuronal markers are observed in PHso2 or in the hermaphrodite PHso1. The switch in gene expression in PHso1 is therefore initiated concomitantly with morphological changes during sexual maturation and in a male-specific manner. Together, these data demonstrate that the male

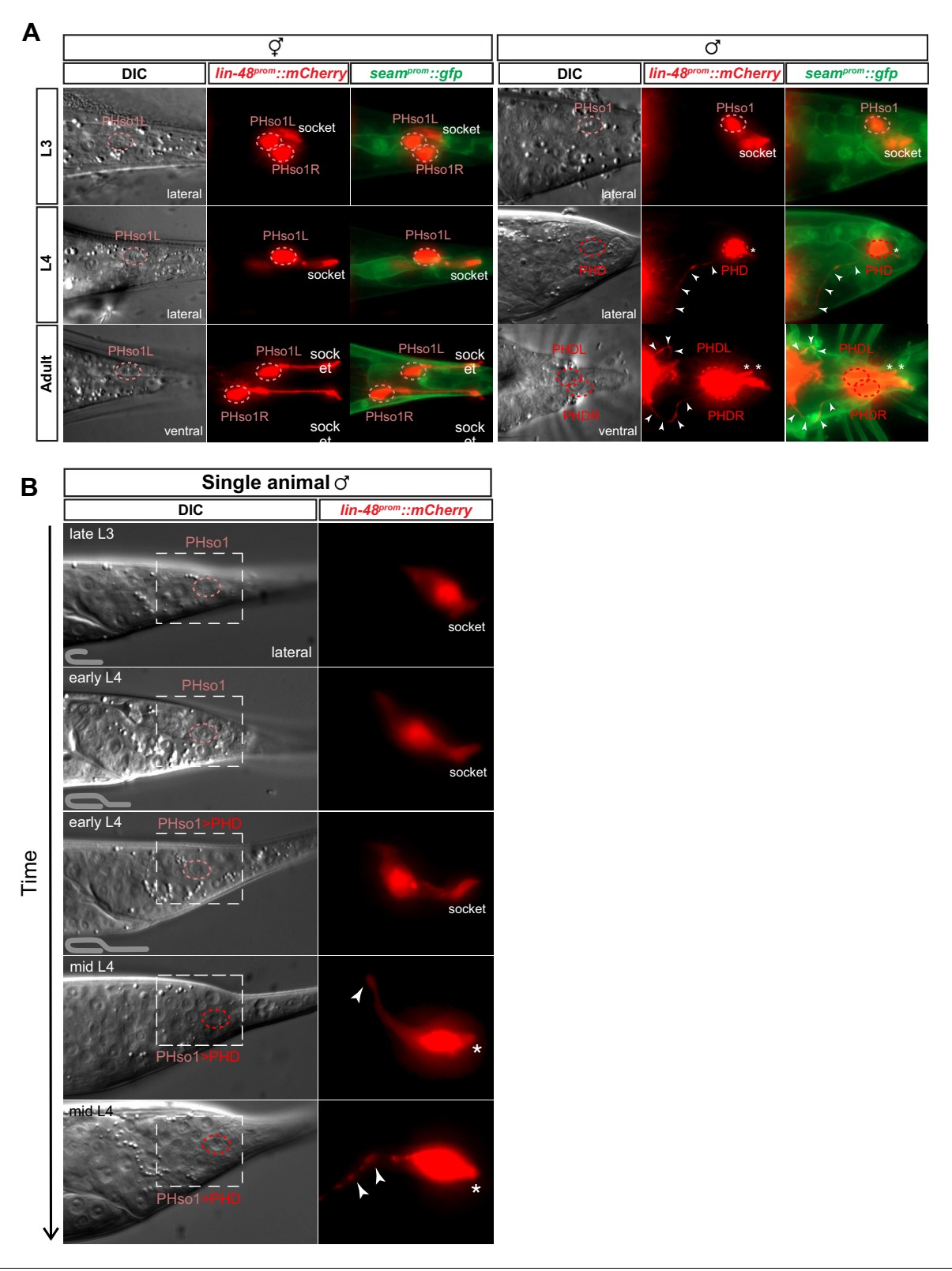

**Figure 2.** The sex-shared PHso1 cells undergo glia-to-neuron morphological changes in males. (A) Expression of *lin-48^prom^::mCherry* and the *seam^prom^:: gfp* (*wrt-2^prom^::gfp*) reporter transgenes in PHso1 of hermaphrodites (left panel) and males (right panel) at the third (L3) and fourth (L4) larval stages and in adults. The images show the morphological transformation of male PHso1 into the PHD neuron during sexual maturation. Arrowheads label the axonal process extending from the PHD into the pre-anal ganglion. Asterisks indicate the dendritic process of PHD. (B) DIC and fluorescent images of a
*Figure 2 continued on next page*

*Figure 2 continued*

time-lapse of PHso1-to-PHD remodelling in an individual male (see Materials and methods). The top two time-points show the late L3 stage after the gonad has looped back and early L4 after the gonad has crossed over itself (see cartoon at the bottom left side of the left panel). The subsequent time-points range from early-to-mid-L4, when the vas deferens has joined with the cloaca, to late mid-L4, when tail-tip retraction is almost complete. The dashed boxes on the DIC images indicate the position of the fluorescent images. Arrowheads indicate the nascent axon.

The online version of this article includes the following figure supplement(s) for figure 2:

**Figure supplement 1.** PHD axon outgrowth.

PHso1 bilateral pair of glial cells transdifferentiate into a novel class of cholinergic and likely peptidergic ciliated neurons, which we have termed phasmid D neurons (PHDs).

## PHso1-to-PHD transdifferentiation is usually but not always direct

We sought to further validate that PHso1 transdifferentiation is direct. Division followed by programmed cell death of one of the daughters is common throughout *C. elegans* embryonic and post-embryonic development (*Sulston and Horvitz, 1977*; *Sulston et al., 1983*). We therefore explored the possibility that PHso1 could divide asymmetrically, to give rise to the PHD neuron and a sister cell that undergoes rapid apoptosis. For this purpose, we analysed mutants of the canonical cell death protein CED-4, homologue to the apoptotic protease activating factor 1 (Apaf-1), required to activate the CED-3 effector caspase (*Ellis and Horvitz, 1986*; *Yuan and Horvitz, 1992*; reviewed in *Conradt et al., 2016*). Strong loss-of-function mutations in *ced-4* result in the survival of somatic cells that normally undergo programmed cell death during development. We observe no statistically significant increase in the frequency of 'survivor cells' in *ced-4(n1162)* mutants compared to controls, when scoring the number of cells per side of each animal. This suggests there is no cell death in the PHso1 lineage (*Figure 3—figure supplement 2A*). Moreover, we never observed an apoptotic cell body in the vicinity of the PHD neuron.

To our surprise, we observed the presence of an extra cell that co-expressed *lin-48^{prom}::tdTomato* and *rab-3^{prom}::yfp* in 13–24% of sides of both *ced-4* mutant and control animals (*Figure 3—figure supplement 2A*). This extra cell had a similar morphology to PHD, was located immediately anterior to PHD and was detected from the early L4 stage onwards. It also expressed all the markers known to be expressed by the PHD neurons (*Figure 3—figure supplement 2B–G*). We realised, however, the percentage occurrence of this extra cell was variable between strains suggesting its presence/absence is background-dependent (*Figure 3—figure supplement 2H*). In the cases where we did observe an extra cell, we wanted to distinguish between the following possibilities: additional expression of the transgenes in a highly similar neuronal subtype, perduring expression of the transgenes from earlier in the lineages of the phasmid sockets or an infrequent division of PHso1. To this end, we performed 5-ethynyl-2′-deoxyuridine (EdU) staining in a *lin-48^{prom}::tdTomato* background. EdU is a thymidine analogue that is incorporated into the replicating DNA of dividing cells. EdU was fed to a synchronised population of animals at late L2-to-early L3 (22 hr after L1 arrest), after the PHso1 cells are born and before they remodel into the PHD neurons, and adult animals were scored. As a positive control, the divison of the bilateral AMso cells that give rise to the MCM neurons were also scored, as it occurs at a similar time as the PHso1-to-PHD transition (*Sammut et al., 2015*). In the majority of cases (79% of sides scored), no EdU signal was detected, indicating that PHso1 mainly transdifferentiates directly into the PHD neuron. In 21% of sides however, EdU signal was detected in two *lin-48^{prom}::tdTomato*-expressing cells per side in the tail of male animals (*Figure 3G*) but never in hermaphrodites (data not shown). Importantly, EdU is never present in the PHD neuron unless observed in two cells on the same side, eliminating the possibility of cell death via non-apoptotic machinery, as is the case of the male linker cell (*Abraham et al., 2007*). In addition, we were able to observe the division of PHso1 in a single animal in a time-lapse of an individual male (*Figure 3—figure supplement 3*).

All together, these observations indicate that PHso1 can sometimes divide symmetrically to give rise to two neurons, which we term PHD1 and PHD2. However, in the vast majority of cases, the PHso1 cells undergo direct remodelling, without wholesale DNA replication or cell division, during which they directly acquire neuronal fate.

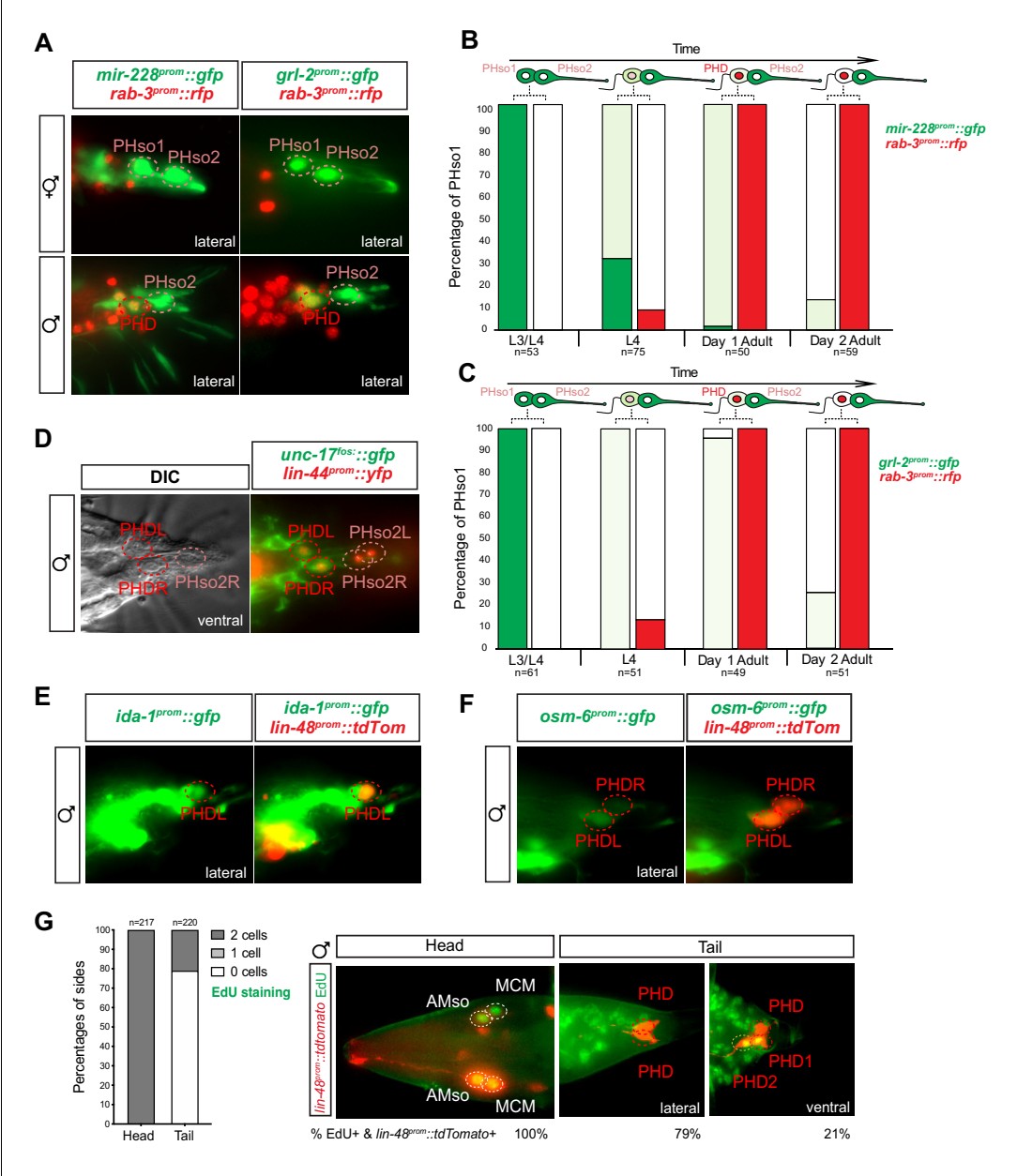

**Figure 3.** The sex-shared PHso1 cells undergo glia-to-neuron molecular changes in males occasionally accompanied by a division. (**A**) Expression of the glial marker reporter transgenes *mir-228prom::gfp* and *grl-2prom::gfp* and pan-neuronal marker *rab-3prom::rfp* in the phasmid socket cells (PHso1 and PHso2) and the PHD neuron in adult male animals. (**B**) Bar chart showing the percentage of PHso1/PHD cells expressing the pan-glial marker reporter transgene *mir-228prom::gfp* and the pan-neuronal marker reporter transgene *rab-3prom::rfp* scored concomitantly in males at different stages of development and at adulthood. Intensity of the *mir-228prom::gfp* reporter transgene in the PHso1/PHD cells was assessed by eye in comparison with PHso2: dark green indicates PHso1/PHD = PHso2, light green indicates PHso1/PHD < PHso2 and white is non-detectable in PHso1. (**C**) As B but for the subtype-specific glial marker *grl-2prom::gfp* scored concomitantly with *rab-3prom::rfp*. (**D**) Expression of the acetylcholine vesicle uploader reporter transgene *unc-17prom::gfp* in the PHD neurons of adult males. The *lin-44prom::yfp* reporter transgene has been coloured red. (**E**) Expression of an *osm-6prom::gfp* reporter transgene in the PHD neurons of adult males, which are co-labelled with a *lin-48prom::tdTomato* transgene. (**F**) Expression of an *ida-1prom::gfp* reporter transgene in the PHD neurons of adult males, which are co-labelled with a *lin-48prom::tdTomato* transgene. (**G**) EdU staining to assess PHso1 division. Left panel: quantification of EdU labelling in cells per side in adult males. The AMso division that gives rise to AMso and MCM cells was scored as a positive control. Right panel: representative images of EdU DNA labelling (green) present in the nuclei of the AMso socket cell and MCM neuron (head), and absent in the PHD neuron (tail), unless two cells per side are observed (PHD1 and PHD2). All cells scored were labelled with a *lin-48prom::tdTomato* transgene.

The online version of this article includes the following source data and figure supplement(s) for figure 3:

**Source data 1.** Scoring data of glial and neuronal fluorescent reporters and EdU labelling in the PHD neurons of *wildtype* animals.

*Figure 3 continued on next page*

*Figure 3 continued*

**Figure supplement 1.** Expression of glial markers is downregulated in the PHso1 of males.
**Figure supplement 2.** PHso1 divides at low frequency in a background-dependent manner.
**Figure supplement 2—source data 1.** Scoring data of PHD background-dependent division.
**Figure supplement 3.** Live division of PHso1 in a single-animal time-lapse.

## Biological sex regulates PHso1-to-PHD transdifferentiation cell-autonomously

We next addressed whether the PHso1-to-PHD cell fate switch is regulated by the genetic sex of the cell rather than the sex of the rest of the animal, in a manner similar to that of several other sexual dimorphisms in *C. elegans* (*Fagan et al., 2018*; *Hilbert and Kim, 2017*; *Lee and Portman, 2007*; *Oren-Suissa et al., 2016*; *Ryan et al., 2014*; *Sammut et al., 2015*; *Weinberg et al., 2018*; *White and Jorgensen, 2012*; *White et al., 2007*). To uncouple the sex of PHso1 from the rest of the animal, we drove expression of *fem-3* in PHso1 (and PHso2) under the *grl-2* promoter. *fem-3* inhibits the expression of *tra-1,* a downstream target of the sex-determination pathway that activates hermaphrodite development and inhibits male development (*Hodgkin, 1987* and reviewed in *Zarkower, 2006*). Therefore, cell-specific expression of a *fem-3* transgene will masculinise a cell in an otherwise hermaphroditic background. We find that *fem-3* expression specifically in the PHso1, and not PHso2, transforms PHso1 into a PHD-like neuron, resulting in the upregulation of *ida-1* and *rab-3* expression and the acquisition of neuronal morphology (*Figure 4A–C*). This indicates that the competence of PHso1 to transdifferentiate into PHD is likely cell-intrinsic and based on genetic sex.

## Factors required for Y-to-PDA transdifferentiation are largely dispensable for PHso1-to-PHD and AMso-to-AMso+MCM transdifferentiation

The first well-described transdifferentiation event in *C. elegans* is the direct conversion of the rectal epithelial cell Y into the motor neuron PDA (*Jarriault et al., 2008*). As with PHso1-to-PHD, Y-to-PDA occurs directly, without a division and in a sex-specific manner. However, Y-to-PDA happens exclusively in hermaphrodites (in males the Y blast cell divides and PDA is generated from the anterior daughter of this division) and during the late L1 stage, much earlier than sexual maturation. In Y-to-PDA, the erasure of the original epithelial fate requires the action of a complex of conserved nuclear factors (SOX-2/Sox, CEH-6/Oct, SEM-4/Sall, and EGL-27/Mta) and the acquisition of the final motor neuron fate involves the cholinergic terminal selector EBF transcription factor UNC-3, in addition to specific combinations of histone modifier complexes (*Kagias et al., 2012*; *Richard et al., 2011*; *Zuryn et al., 2014*). We investigated how broadly the molecules involved in the initiation of Y-to-PDA transdifferentiation could be acting in *C. elegans*, as we hypothesised that the factors involved in the re-specification would depend on the specific terminal neuronal cell fate. Hence, we analysed loss-of-function mutants for *sem-4, egl-27*, and *sox-2* in PHso1-to-PHD transdifferentiation, and in the previously described AMso-to-AMso+MCM cell fate switch (*Sammut et al., 2015*).

We observed no statistically significant defect in the number and cell identity of MCM and PHD neurons in *sem-4(n1971)* null mutants (*Figure 5A*). In *egl-27(ok1671)* strong loss-of-function mutants, although a significant defect is observed in the number of *lin-48^{prom}::tdTomato*-positive MCM neurons, normal *rab-3^{prom}::yfp* expression suggests a defect in *lin-48::tdTomato* reporter transgene expression levels but not in the division of the AMso or in the acquisition of neuronal fate by the MCM. No *lin-48^{prom}::tdTomato* expression is ever observed in the region of PHso1/PHD in *egl-27* mutants at any stage or in either sex (*Figure 5A* and data not shown). This is consistent with previous results that show that *egl-27* affects the division pattern of the T lineage and consequently the production of PHso1 (*Herman et al., 1999*). This early role of *egl-27* in the PHso1 lineage precludes its study in the generation of the PHD neuron using loss-of-function mutants.

As *sox-2* null mutations cause L1 lethal larval arrest, we used two parallel strategies to investigate the role of *sox-2* in PHso1-to-PHD and AMso-to-AMso+MCM: RNAi knock-down and analysis of a rescued *sox-2* null allele. For the former, we made use of the RNAi-sensitised strain *nre-1(hd20) lin-15B(hd126)* (*Schmitz et al., 2007*). No statistically significant defect is observed in the PHD neuron in *sox-2* RNAi-treated animals (*Figure 5B*). An increase in the number of *rab-3^{prom}::yfp*-expressing

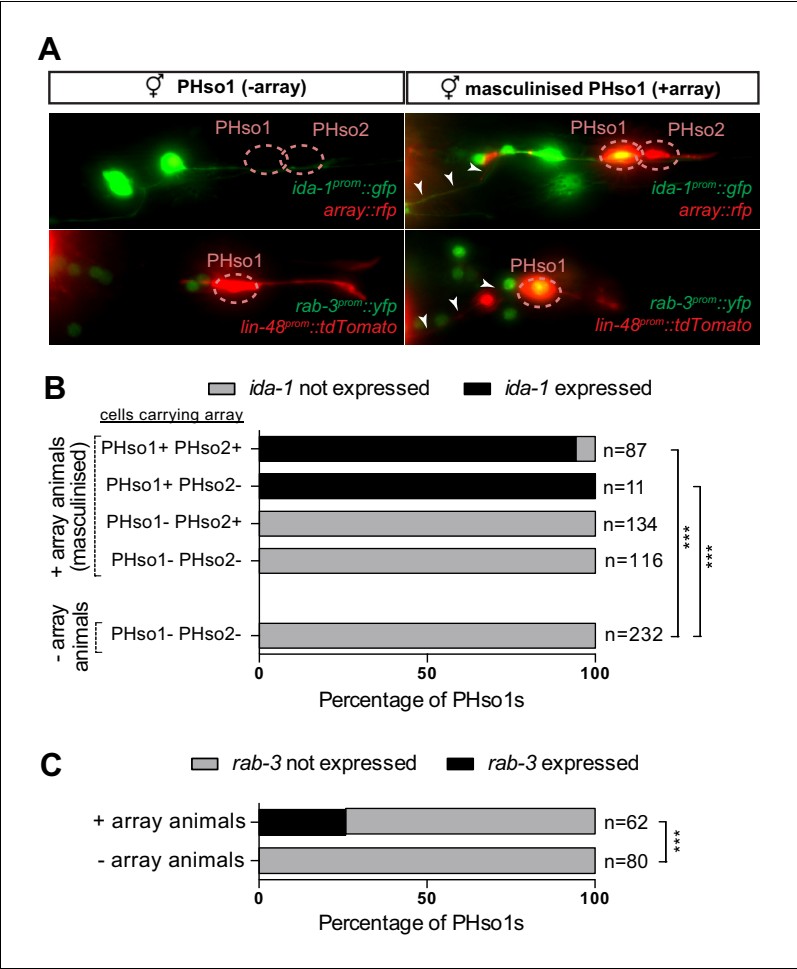

**Figure 4.** PHso1-to-PHD plasticity is intrinsically regulated. (A) Expression of the *ida-1^prom^::gfp* and *rab-3^prom^::yfp* reporter transgenes in adult hermaphrodites carrying the masculinising array *grl-2^prom^::fem-3::mCherry* in PHso1 (right panel) and in non-array-carrying hermaphrodites (left panel). (B) Bar chart showing the percentage of PHso1 and PHso2 cells expressing the *ida-1^prom^::gfp* reporter transgenes in adult hermaphrodites carrying or not carrying the masculinising *grl-2^prom^::fem-3::mCherry* array. Of note, the *grl-2* promoter fragment is also expressed in the AMso, excretory pore and excretory duct cells in the head (**Hao et al., 2006**). Fisher's exact test was used to compare all categories between genotypes and only statistically significant differences from the non-array carrying animals are indicated (*p≤0.05, **p≤0.01, ***p≤0.001). (C) Bar chart showing the percentage of PHso1 cells expressing the *rab-3^prom^::yfp* reporter transgene in adult hermaphrodites carrying or not carrying the masculinising *grl-2^prom^::fem-3::mCherry* array. Fisher's exact test was used to compare all categories between genotypes and only statistically significant differences from non-array carrying animals are indicated (*p≤0.05, **p≤0.01, ***p≤0.001).

The online version of this article includes the following source data for figure 4:

**Source data 1.** Scoring data of PHso1 masculinised cells in hermaphrodite worms.

MCM cells was detected (*Figure 5B*) but analysis of *lin-48^prom^::tdTomato* expression revealed that the extra *rab-3* positive cells derive from ectopic AMso cells that divide efficiently to produce MCM neurons (*Figure 5—figure supplement 1A*). Interestingly, the ectopic MCM cells recapitulate normal development as they not only express the pan-neuronal marker *rab-3^prom^::yfp*, but also the neuron subtype marker *ida-1^prom^::gfp* (*Figure 5—figure supplement 1B*). Moreover, although ectopic AMso cells are observed in both sexes, they exclusively divide in males and not in hermaphrodites (*Figure 5—figure supplement 1A*). This, together with the fact that ectopic *lin-48*-positive cells are induced in RNAi-treated animals only in the F1 generation (and not the P0 generation – see

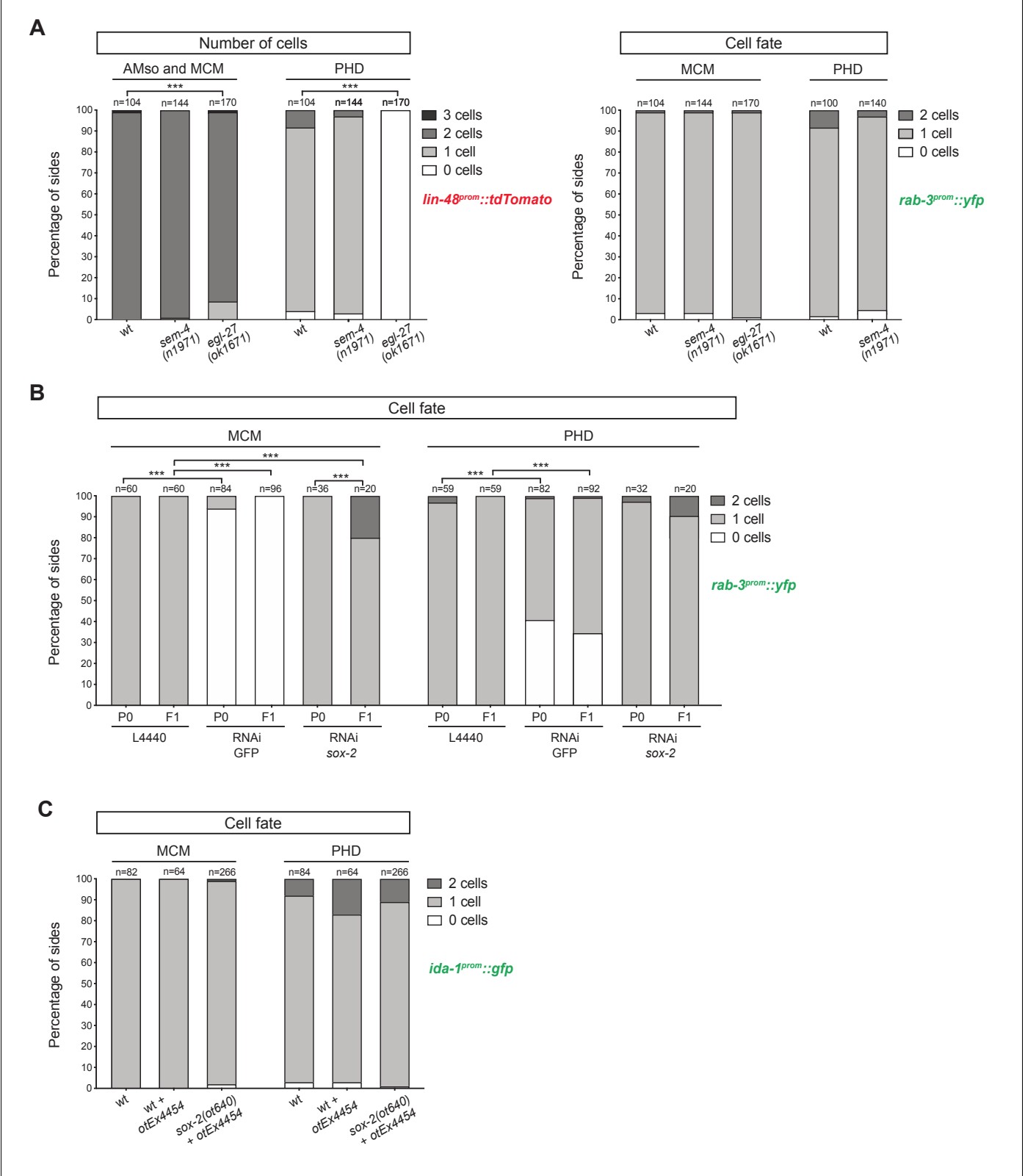

**Figure 5.** Factors required for Y-to-PDA transdifferentiation are largely dispensable for PHso1-to-PHD and AMso-to-AMso+MCM transdifferentiation. (**A**) Bar chart showing the percentage of AMso, MCM, and PHD cells expressing the *lin-48^prom^::tdTomato* transgene (*drpIs3;* left panel) and the pan-neuronal marker *rab-3^prom^::yfp* reporter transgene (*otIs291;* right panel), in *sem-4(n1971)* putative null and *egl-27(ok1670)* strong loss-of-function mutant animals. The presence and morphology of cells was assessed by *lin-48^prom^::tdTomato* that is expressed, in the head, in the MCM mother (the AMso)

*Figure 5 continued on next page*

*Figure 5 continued*

and retained in the AMso and MCM daughters after the division (two cells per side). In the tail, *lin-48^{prom}::tdTomato* is expressed in PHso1 before the cell remodelling and after, in the PHD neuron (one cell per side unless PHD1 and PHD2 are observed). Cells per side were scored in male animals from late L4 to adult stages. Neuronal identity was assessed by *rab-3^{prom}::yfp*. For the PHD neuron *rab-3^{prom}::yfp* was scored concomitantly with *lin-48^{prom}::tdTomato* due to the high number of *rab-3^{prom}::yfp*-expressing neurons in the tail. Fisher's exact test was used to compare all categories between genotypes and only statistically significant differences from the *wildtype* phenotype are indicated (*p≤0.05, **p≤0.01, ***p≤0.001). Of note, 4/7 cells lacking *rab-3^{prom}::yfp* expression in the tail of *sem-4* mutants retained a socket morphology, which is never observed in the cells lacking the reporter in the control strain. This could suggest a block to the initiation of transdifferentiation in PHso1 or that the severe morphological defects of male tails in these mutant animals impair the remodelling process. (B) Bar chart showing the percentage of MCM and PHD neurons expressing *rab-3^{prom}::yfp* after RNAi-knockdown of *sox-2*. L4440 empty vector was used as a negative control and GFP RNAi-knockdown as a positive control. Fisher's exact test was used to compare all categories between genotypes and statistically significant differences from the *wildtype* phenotype are indicated (*p≤0.05, **p≤0.01, ***p≤0.001). (C) Bar chart showing the percentage of MCM and PHD neurons expressing the *ida-1^{prom}::gfp* neuron subtype marker in *sox-2 (ot460)* null mutant animals rescued for lethality with a *sox-2* fosmid-based extrachromosomal array: (*otEx4454[sox-2(fosmid)::mCherry + elt-2^{prom}::DsRed]*). A mixed population of of *sox-2* mutant (mosaic or non-rescued) and *wildtype* (mutant-rescued) cells were scored. No statistical difference is observed between any of the groups using Fisher's exact test.

The online version of this article includes the following source data and figure supplement(s) for figure 5:

**Source data 1.** Scoring data of glial and neuronal fluorescent reporters in the PHD neurons of *sem-4*, *egl-27*, and *sox-2* mutant animals.
**Figure supplement 1.** Factors required for Y-to-PDA transdifferentiation are largely dispensable for PHso1-to-PHD and AMso-to-AMso+ MCM transdifferentiation – additional data.
**Figure supplement 1—source data 1.** Additional scoring data of glial and neuronal fluorescent reporters in the PHD neurons of *sem-4*, *egl-27*, and *sox-2* mutant animals.

---

Materials and methods), suggests an early role of *sox-2* in an unknown lineage to supress the production of supernumerary AMso cells.

We additionally analysed a strain containing a *sox-2(ot640)* null allele that was not available at the time of Y-to-PDA characterisation (*Vidal et al., 2015*). To bypass lethality, this strain contains an extrachromosomal *sox-2* fosmid-based rescue array. The presence of this array in individual worms can be tracked by a fluorescent co-injection transgene, *elt-2^{prom}::DsRed*. Within an individual animal the array is inherited in a mosaic fashion and this is validated by the presence of dead embryos positive for the array. As with RNAi-knockdown, no statistically significant defect is observed in the PHD neuron (*Figure 5C*). In the head, one ectopic *ida-1^{prom}::gfp*-expressing MCM neuron is overserved in 3/266 sides per animal, coinciding with an extra AMso (*Figure 5C* and *Figure 5—figure supplement 1C*) and phenocopying the F1 RNAi results. However, four out of 269 total *lin-48^{prom}::tdTomato* expressing MCM cells lack *ida-1^{prom}::gfp* expression (*Figure 5C* and *Figure 5—figure supplement 1C*), suggesting a possible minor role of *sox-2* in AMso-to-AMso+MCM, not detected in the RNAi-knockdown experiments.

Mutants in the same genes implicated in Y-to-PDA transdifferentiation were employed in this work and with the exception of *sox-2*, identical alleles were used. However, while the Y cell completely fails to transdifferentiate into PDA in these mutants (70–100% of the cases - *Kagias et al., 2012*), the PHso1-to-PHD and AMso-to-AMso+MCM cell fate switches show no or only very mild defects. Altogether these results suggest that different molecular mechanisms from the ones regulating Y-to-PDA must govern the PHso1-to-PHD and AMso-to-AMso+MCM cell fate switches.

## PHDs are sensory neurons of male-specific copulation circuits

To further establish the neuronal characteristics of PHDs, we examined their synapses and ultrastructure, and mapped their full wiring diagram. We identified a ~1 kb promoter region that specifically drives the expression of *oig-8* in the PHD neurons. Expression of a *rab-3* translational fusion under the control of this *oig-8* promoter reveals that PHDs form synapses in the pre-anal ganglion, where the synaptic vesicle associated mCherry-tagged RAB-3 protein can be observed (*Figure 6A*). At the ultrastructural level, we observed both synaptic and dense-core vesicles in PHD, proximal to the cell body (*Figure 6B*). Ultrastructural analysis of the PHD dendrite in seven different animals independently confirms John Sulston's original observation on the presence of cilia, including the basal body and axonemes (*Sulston et al., 1980*). The basal body of PHD lies just dorsal and anterior to the phasmid sheath channel containing the PHA and PHB cilia. It has 2, 1, or 0 short cilia extending

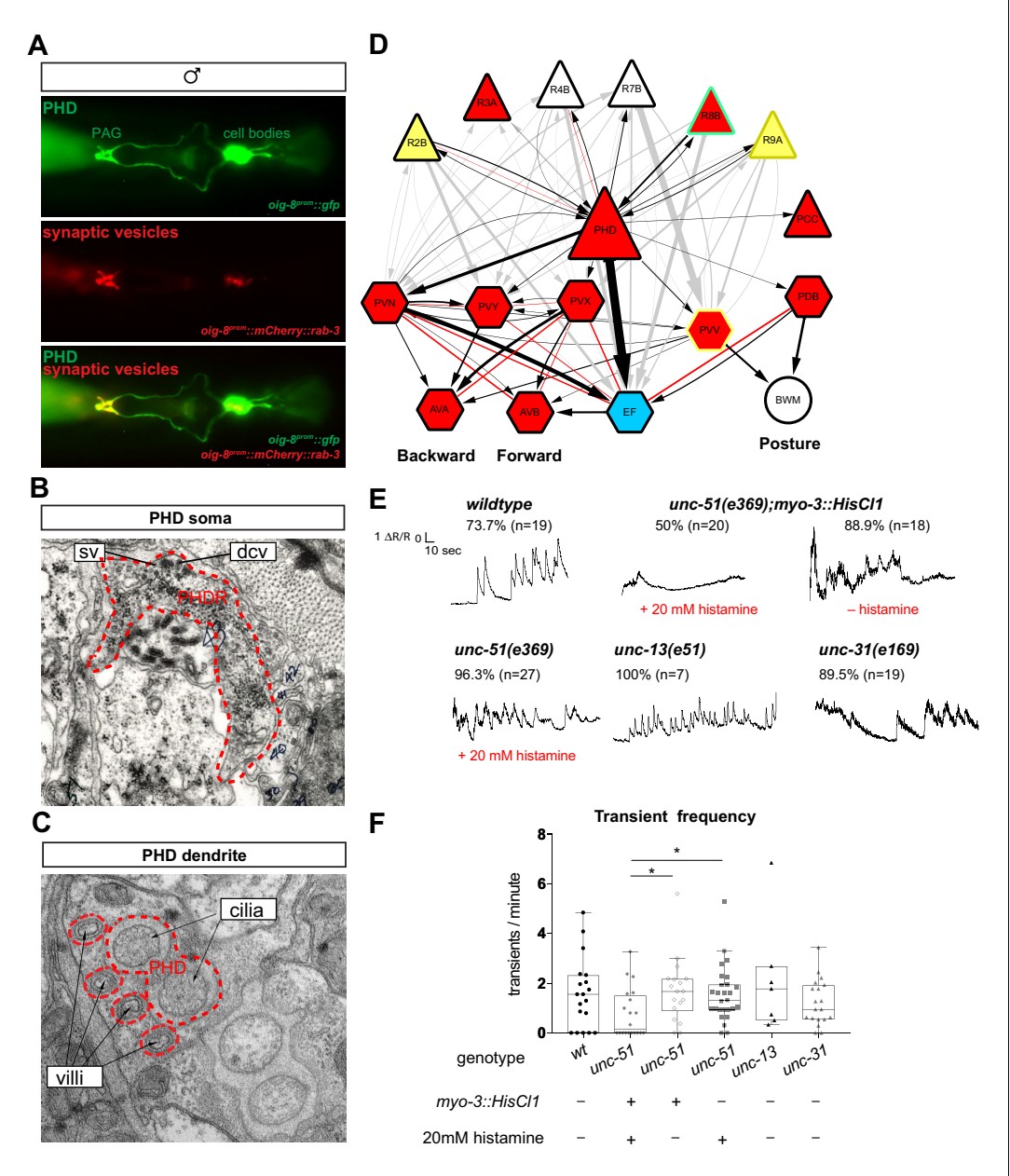

**Figure 6.** The PHDs are putative proprioceptive neurons of male-specific copulation circuits. (**A**) Expression of the *oig-8[prom]::mCherry::rab-3* and *oig-8[prom]::gfp* reporter in the PHD neurons of adult males. The synapses made by the PHDs in the pre-anal ganglion (PAG) can be observed (ventral view). (**B**) Electron micrographs of the soma of a PHD neuron of an adult male. sv, synaptic vesicle; dcv, dense-core vesicles; PHDR, right PHD neuron. (**C**) As B for a PHD dendrite. (**D**) Diagram depicting the connectivity of the PHD neurons with their main pre-synaptic inputs (ray neurons) and post-synaptic targets. The connections between the ray neurons (RnA/B) and their post-synaptic targets independently of PHD are indicated in grey. Arrows and red lines indicate chemical and electrical synaptic connections, respectively. The thickness of the arrows is proportional to the anatomical strength of their connections (# serial sections). Neurons are colour-coded according to their neurotransmitter: red, cholinergic; yellow, glutamatergic; dark yellow, dopaminergic; blue, GABAergic; green, serotonergic; white, orphan. Note that some neurons (R8B, R9A, PVV) express more than one neurotransmitter. (**E**) Example traces showing PHD activity as normalised GCaMP/RFP fluorescence ratio in restrained animals. Traces are shown for a *wildtype* male, an *unc-51(e359)* mutant with and without a histamine-inducible silencing transgene in muscle (*myo-3[prom]::HisCl1*), a mutant in synaptic transmission (*unc-13*), and a mutant in dense-core vesicle exocytosis (*unc-31*, CADPS/CAPS). The proportion of traces where calcium peaks were identified is indicated for each genotype and treatment. n = number of neurons imaged. (**F**) Plots of frequency values of calcium transients per neuron. Dots represent

*Figure 6 continued on next page*

*Figure 6 continued*

individual neurons imaged. Tukey box-and-whisker plots indicate the interquartile ranges and median. *p<0.05; One-way ANOVA with multiple comparisons. Two groups were compared: *unc-51* genotypes and treatments and another group including *wt, unc-13,* and *unc-31*. Only statistically significant comparisons are indicated.

The online version of this article includes the following source data and figure supplement(s) for figure 6:

**Source data 1.** Value of calcium transients elicited by PHD in different genetic backgrounds.
**Figure supplement 1.** Ultrastructure of the male phasmid sensilla.

within the phasmid sheath very close to PHA and PHB cilia, but within a separate sheath channel. Interestingly, the PHD cilia can lie in variable positions relative to the PHA and PHB cilia: medial, lateral, dorsal, or ventral. In addition, we observe that the dendrite is more elaborate than previously described and has a number of unciliated finger-like villi within the phasmid sheath cell, proximal to the basal body (*Figure 6C* and *Figure 6—figure supplement 1*). Through reconstruction of serial electron micrographs all the synaptic partners of PHD were identified (*Supplementary file 1*; *Jarrell et al., 2012*). PHD neurons project from the dorsolateral lumbar ganglia anteriorly and then ventrally along the posterior lumbar commissure and into the pre-anal ganglion. In the pre-anal ganglion, these establish chemical synapses and gap junctions with sensory neurons and interneurons, most of which are male-specific (*Figure 6D*). Previously, the axonal process of PHD was attributed to the R8B ray neuron (*Jarrell et al., 2012*). A re-examination of the lumbar commissure allowed us to disambiguate PHD's axon. PHDs receive synaptic input from male-specific ray sensory neurons involved in the initiation of the mating sequence in response to mate contact (*Barr and Sternberg, 1999*; *Liu and Sternberg, 1995*; *Koo et al., 2011*). The main PHD output is to the male-specific EF interneurons (35.4% of chemical synapses), both directly and through their second major post-synaptic target, the sex-shared PVN interneuron. The EF interneurons are GABAergic (*Serrano-Saiz et al., 2017b*) and synapse onto the AVB pre-motor interneurons which drive forward locomotion. Other PHD outputs include the PVV (male-specific) and PDB (sex-shared but highly dimorphic in connectivity; *Jarrell et al., 2012*; *Cook et al., 2019*) interneurons, which synapse onto body-wall muscle, and the cholinergic male-specific interneurons PVY and PVX whose output is to the AVA pre-motor interneurons, which drive backward locomotion. PVN, PVY, and PVX form disynaptic feed-forward triplet motifs that connect PHD strongly to the locomotion circuit interneurons AVB and AVA. The pattern of connectivity of PHD suggests a possible role in male mating behaviour.

## PHDs are putative proprioceptive neurons

We next sought to establish the function of the PHD neurons. We began by asking what sensory stimuli might activate them. To monitor neuronal activity, we co-expressed GCaMP6f and RFP in the PHDs and performed ratiometric measurements of fluorescence. We imaged calcium transients in restrained animals glued to a slide without anaesthetic. To our surprise, without applying any exogenous stimulation, we observed intermittent calcium transients in the PHDs of *wildtype* animals, every 30 s on average (*Figure 6E–F* and *Video 1*). We observed calcium transients in 73% of the neurons imaged and no peaks could be identified in the remaining traces (n = 19). Since in restrained animals there are small tail and spicule movements due to the defecation cycle and sporadic muscle contractions, one explanation could be that PHDs

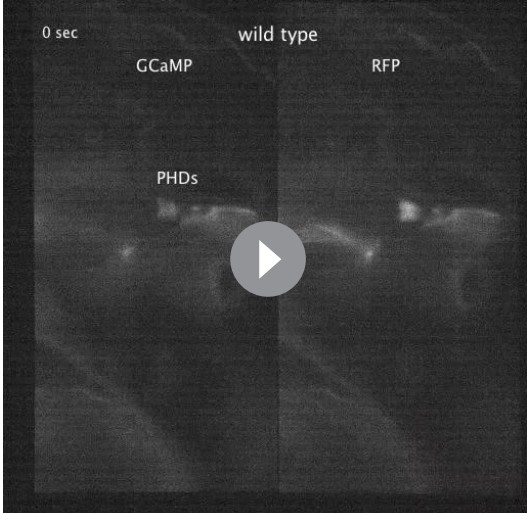

**Video 1.** Imaging of neuronal activity in PHD neurons with GCaMP6f (left channel) and RFP (right channel) in restrained animals: *wildtype* male. Animals are expressing an *oig-8$^{prom}$::GCaMP6f::sl2::rfp* transgene. Videos play at 100 fps (recorded at 20 fps).
https://elifesciences.org/articles/48361#video1

may be proprioceptive. We reasoned that if PHD activity resulted from internal tissue deformation caused by muscle contractions, inhibition of muscle activity should eliminate or reduce calcium transients. To inhibit muscle contractions, we generated transgenic worms in which muscles can be silenced in an inducible manner through the expression of the *Drosophila* histamine-gated chloride channel HisCl1 (*Pokala et al., 2014*), under the *myo-3* promoter. To increase the efficiency of muscle silencing, we introduced this transgene into an *unc-51(e369)* mutant background that renders animals lethargic. *unc-51* encodes a serine/threonine kinase required for axon guidance and is expressed in motor neurons and body-wall muscle (*Ogura et al., 1997*). Histamine-treated *myo-3^prom::HisCl1;unc-51(e369)* animals were highly immobile and the frequency of calcium transients in the PHDs was strongly reduced (*Figure 6E–F* and *Videos 2* and *3*). Transients were completely eliminated in half of the neurons (*Figure 6F*). The reduction in transient frequency was specific to the silencing of the muscles because in histamine-treated *unc-51(e369)* and histamine non-treated *myo-3^prom::HisCl1;unc-51(e369)* control animals, frequency was similar to that of *wildtype* animals (*Figure 6E,F* and *Video 4*).

Although PHDs are sensory neurons, they do receive some small synaptic input from other neurons (*Figure 6D*) and therefore the activity observed in restrained animals in response to muscle contractions could arise either directly through PHD sensory input or indirectly through pre-synaptic neurons. To test this, we imaged PHD activity in *unc-13(e51)* mutant males with impaired synaptic neurotransmission (*Miller et al., 1996*) and in *unc-31(e169)* mutant males with impaired dense-core vesicle secretion (*Speese et al., 2007*). In both these mutants, PHD calcium transients persisted at similar frequencies to those in *wildtype* animals (*Figure 6E and F*). This suggests that PHD activity does not require chemical input from the network. However, we cannot completely rule out that residual neurotransmission in the hypomorphic *unc-13(e51)* mutants may be sufficient to trigger *wildtype* levels of PHD activity in response to muscle contractions. Together, these results suggest that PHDs may respond directly to internal cues arising from muscle contractions, and that they may be proprioceptive neurons. PHDs may sense internal tissue deformations through their elaborate ciliated dendrites, which are deeply encased in the phasmid sheath and not exposed to the outside environment.

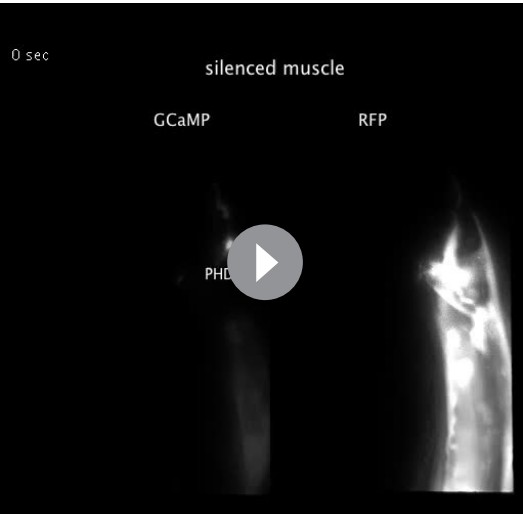

**Video 2.** Imaging of neuronal activity in PHD neurons with GCaMP6f (left channel) and RFP (right channel) in restrained animals: *unc-51(e359)* male expressing a histamine-inducible silencing transgene in muscle (*myo-3^prom::HisCl1::mCherry*) and treated with 20mM histamine. Animals are expressing an *oig-8^prom::GCaMP6f::sl2::rfp* transgene. Videos play at 100 fps (recorded at 20 fps).
https://elifesciences.org/articles/48361#video2

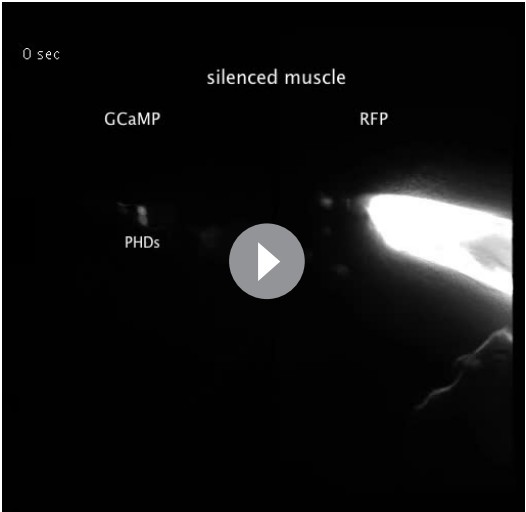

**Video 3.** Imaging of neuronal activity in PHD neurons with GCaMP6f (left channel) and RFP (right channel) in restrained animals: *unc-51(e359)* male expressing a histamine-inducible silencing transgene in muscle (*myo-3^prom::HisCl1::mCherry*) and treated with 20mM histamine. Animals are expressing an *oig-8^prom::GCaMP6f::sl2::rfp* transgene. Videos play at 100 fps (recorded at 20 fps).
https://elifesciences.org/articles/48361#video3

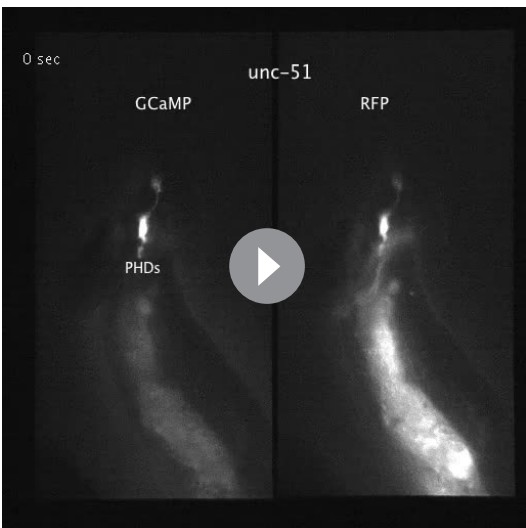

**Video 4.** Imaging of neuronal activity in PHD neurons with GCaMP6f (left channel) and RFP (right channel) in restrained animals: *unc-51(e359)* male treated with 20mM histamine. Animals are expressing an *oig-8^prom^::GCaMP6f::sl2::rfp* transgene. Videos play at 100 fps (recorded at 20 fps).
https://elifesciences.org/articles/48361#video4

## A novel readjustment step during male mating

The connectivity of PHD to male-specific neurons in the tail suggests that they may play a role in male reproductive behaviours controlled by tail circuits. In *C. elegans,* these include food-leaving behaviour, an exploratory strategy to search for mates (*Lipton et al., 2004*; *Barrios et al., 2012*; *Barrios et al., 2008*), and mating behaviour. Mating consists of a temporal sequence of discrete behavioural steps: response (to mate contact); scanning (backward locomotion while scanning the mate's body during vulva search); turning (at the end of the mate's body to continue vulva search); location of vulva (stop at vulva); spicule insertion (intromission); and sperm transfer (reviewed in *Barr et al., 2018*; *Figure 7*). The coordinated execution of these mating steps relies on sensory cues from the mate (*Barr and Sternberg, 1999*) and also, presumably, proprioceptive inputs within the male's copulation circuit as well (*Sulston et al., 1980* and reviewed in *García, 2014*). This sensory information guides the male to either initiate the next step of the sequence or to reattempt the current, unsuccessful step through readjustment of movement and/or posture (reviewed in *Barr et al., 2018*).

During our behavioural analysis of *wildtype* males, we identified a novel readjustment movement that has not been previously described which we have termed the 'Molina manoeuvre'. This movement occurs when the male has been trying to insert its spicules in the mate's vulva for a period of time without success and subsequently loses vulva apposition. Unsuccessful spicule insertion attempts occurred in 65% of males, resulting in either a long displacement from the vulva, which led to the re-initiation of the scanning sequence, or in a small displacement (less than two tail-tip length) from the vulva (98% of vulva losses), as previously described (*Correa et al., 2012*). We find that these small displacements from the vulva lead either to local shifts to relocate the vulva (68% of total vulva losses) or to a Molina manoeuvre (32% of vulva losses) (*Figure 7*). The Molina manoeuvre consists of the initiation of forward locomotion along the mate's body away from the vulva to the end of the mate's body (or at least two male tail-tips distance away from the vulva), at which point the male tail acquires a deeply arched posture, followed by a return to the vulva with backward locomotion along the same route (*Video 5*). Males perform this movement towards either end (head or tail) of the mate as a smooth, continuous sequence. Although we used paralysed *unc-51* mutant hermaphrodites to aid our mating analysis, males also perform this readjustment movement with moving *wildtype* mates (*Video 6*).

## The PHD neurons are required for coordinated backward locomotion and effective intromission during mating

To test the hypothesis that PHDs regulate reproductive behaviours, we performed behavioural tests of intact and PHD-ablated males and functional imaging of PHD activity in freely behaving males during mating. Target specificity of the laser ablations was confirmed 1-day after ablation by absence of fluorescence returning from the transgene used to label the PHD neurons (*Figure 7—figure supplement 1A–C*). We found no defects in food-leaving behaviour, response to mate contact, turning, location of vulva, or spicule insertion (*Figure 7A,B,D,E and H*). However, these experiments revealed a role for PHDs in initiation and/or maintenance of backward locomotion during scanning and during the Molina manoeuvre. PHD-ablated males often switched their direction of locomotion during scanning, performing fewer continuous backward scans during vulva search compared to

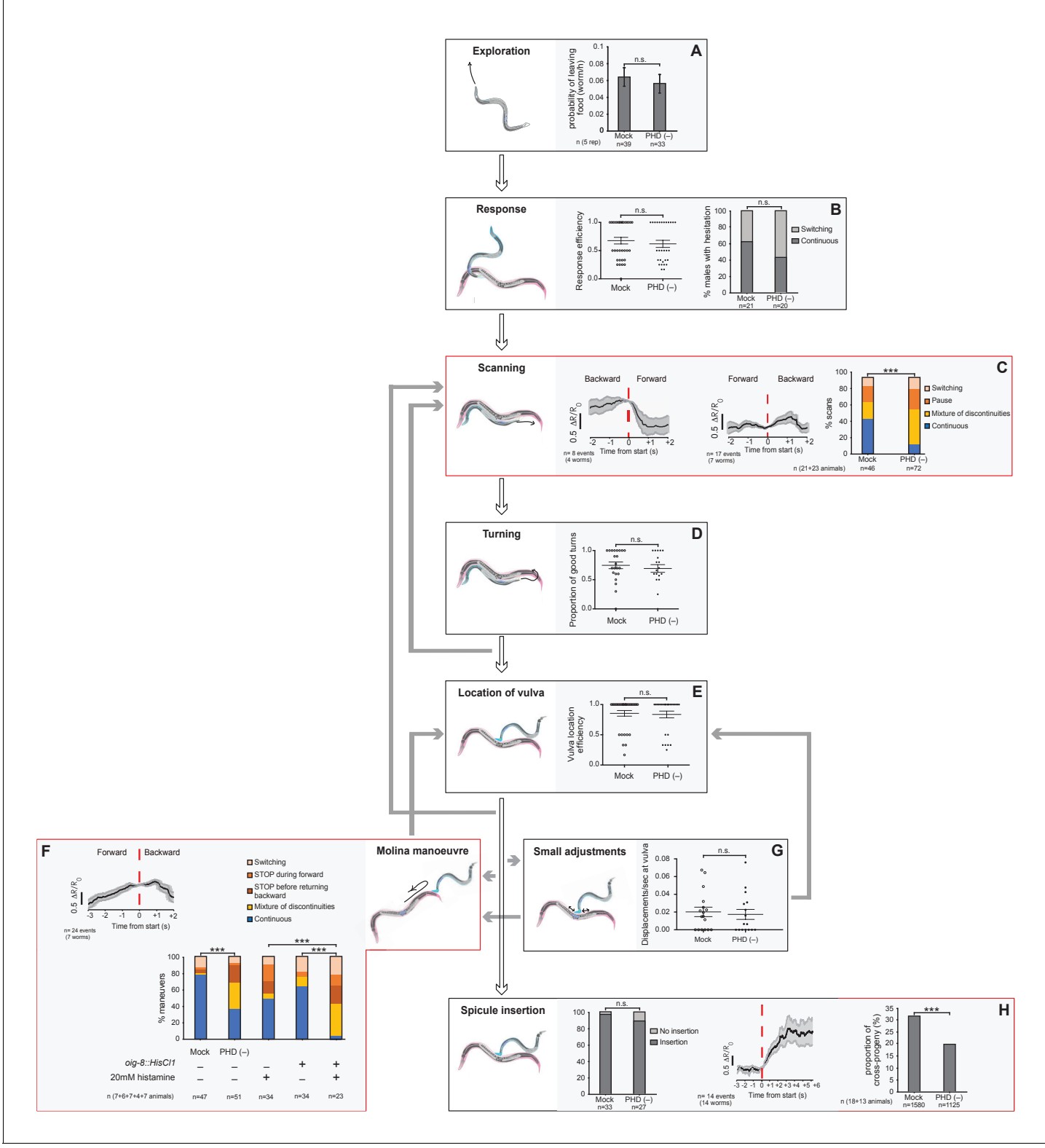

**Figure 7.** The PHD neurons are required for coordinated backward locomotion and effective intromission during mating. Diagram depicting the steps of reproductive behaviours controlled by the male tail circuits. The steps affected by PHD ablation are highlighted in red. Intact (mock) and ablated males carried either an *oig-8^prom^::gfp* or an *unc-17^prom^::gfp* transgene to identify the PHD neurons for ablation. In F, males carrying an *oig-8^prom^::HisCl1* transgene were used to silence PHD neurons acutely. White arrows indicate the transitions between the mating sequence. Grey arrows indicate the corrective transitions that males perform when they fail to attain the subsequent goal. The corrective transitions that occur upon failure of spicule

*Figure 7 continued on next page*

*Figure 7 continued*

insertion attempts (between location of vulva and spicule insertion) are always preceded by a displacement from the vulva (not depicted). Behavioural analysis in intact and PHD-ablated males are shown for each step. Calcium imaging in PHD neurons is shown for steps C, F, and H. The black trace shows PHD activity as normalised GCaMP/RFP fluorescence ratio changes averaged for several events and phase locked (red dotted line) to the switch in the direction of locomotion (**C and F**) or to the start of spicule insertion (**H**). The grey shadow shows S.E.M. (**A**) Male exploratory behaviour measured as $P_L$ values (probability of leaving food per worm per hour). n, number of males tested. Maximum likelihood statistical analysis was used to compare $P_L$ values. n.s., no statistically significant difference, $p \geq 0.05$; error bars, S.E.M. (**B**) Response efficiency, measured as 1/number of contacts with a mate before responding; and hesitation, measured as a switch in direction of locomotion during response. (**C**) Scanning locomotion during vulva search. Categories: switching (change in the direction of locomotion from backward to forward); pause (stopping during backward scanning); mixture (scans with switches and pauses); continuous (uninterrupted backward movement along the mate's body). $p<0.001$ ($\chi^2$ test of continuous and discontinuous scans); n = number of events. (**D**) Turning, measured as proportion of good turns per male. (**E**) Vulva location efficiency measured as 1/number of vulva encounters before stopping. (**F**) Proportion of continuous and discontinuous Molina manoeuvres. Categories: switching (a brief change in the direction of movement either during forward locomotion away from the vulva or during backward locomotion returning to the vulva); STOP during forward (stopping during forward movement away from the vulva and then continuing in the same direction); STOP before returning backward (stopping at the transition between forward movement away from the vulva and returning backwards to the vulva); mixture (manoeuvres that displayed more than one of the discontinuities described above); continuous (smooth movement forward away from the vulva and return backwards to the vulva without stopping or switching in between). $p<0.001$ ($\chi^2$ test of continuous and discontinuous manoeuvres); n = number of events. (**G**) Number of displacements away from the vulva per unit of time spent at vulva. (**H**) Left bar chart, proportion of males able to insert their spicules; n.s., no statistically significant difference, $p \geq 0.05$ ($\chi^2$ test); n = animals tested. Right bar chart, sperm transfer efficiency measured as percentage of cross-progeny; $p<0.001$ ($\chi^2$ test); n = total progeny. For B, D, E, and G, bar and dots represent mean and individual animal values, respectively; error bars, S.E.M. n.s., no statistically significant difference, $p \geq 0.05$ (Mann-Whitney U test). Worm cartoons were modified with permission from original drawings by Rene García.
© 2018, Genetics Society of America. Figure 7 contains modified versions of the male and hermaphrodite worm cartoons from *Barr et al., 2018* used with permission. They are not covered by the CC-BY 4.0 licence and further reproduction of this panel would need permission from the copyright holder.
The online version of this article includes the following source data and figure supplement(s) for figure 7:

**Source data 1.** Analysis of mate searching, fertility, and male-mating behaviours and measurements of PHD neuronal activity during mating in *wildtype* and PHD-ablated animals.
**Figure supplement 1.** PHD neuron ablation control.
**Figure supplement 2.** Some ectopic prodding occurs during discontinuous Molina manoeuvres and during pauses while scanning.
**Figure supplement 2—source data 1.** Scoring of ectopic prodding events during mating in *wildtype* and PHD-ablated animals.

intact males (*Figure 7C*). Consistent with this, we observed higher activity (i.e. Ca²⁺ levels) in PHD during backward locomotion than during forward locomotion while scanning (*Figure 7C*). PHD activity also peaked just after the switch to backward locomotion during Molina manoeuvres, when the tail tip displays an acute bent posture, and PHD-ablated males performed defective, discontinuous manoeuvres, often stopping at the transition from forward to backward locomotion to return to the vulva (*Figure 7F* and *Video 7*). Importantly, we did not find significant differences between intact and PHD-ablated males in number of events that trigger the initiation of manoeuvres (*Figure 7G*), indicating that PHDs are specifically required for coordinated locomotion during the manoeuvre itself, at the transition point to backward locomotion. Similar locomotion defects resulting in discontinuous manoeuvres were observed when PHD neurons were acutely silenced with an inducible chloride channel transgene (*oig-8ᵖʳᵒᵐ::HisCl1::sl2::rfp)* (*Figure 7G*). Together, these data demonstrate that without intact PHD neurons, backward movement along the mating

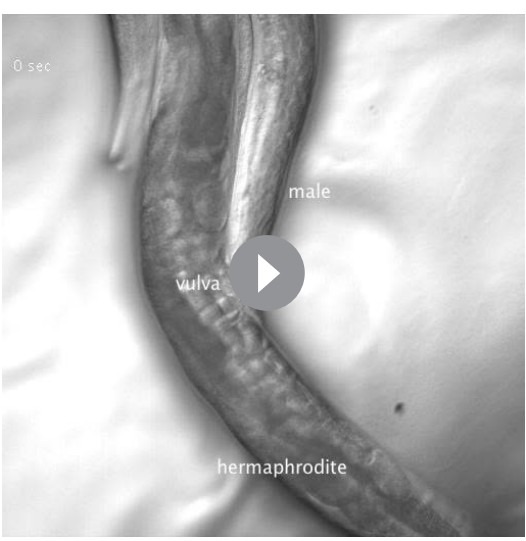

**Video 5.** Males performing Molina manoeuvres during mating: *wildtype* male performing a Molina manoeuvre during mating with a paralysed *unc-51 (e359)* hermaphrodite. Videos are played at 40 fps (sped up x2).
https://elifesciences.org/articles/48361#video5

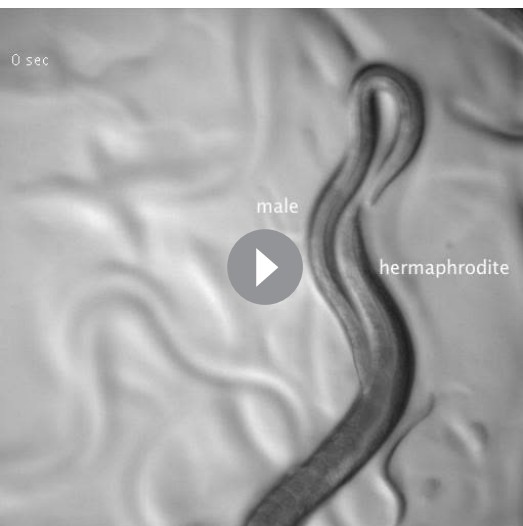

**Video 6.** Males performing Molina manoeuvres during mating: *wildtype* male performing a Molina manoeuvre during mating with a *wildtype* hermaphrodite. Videos are played at 40 fps (sped up x2).

https://elifesciences.org/articles/48361#video6

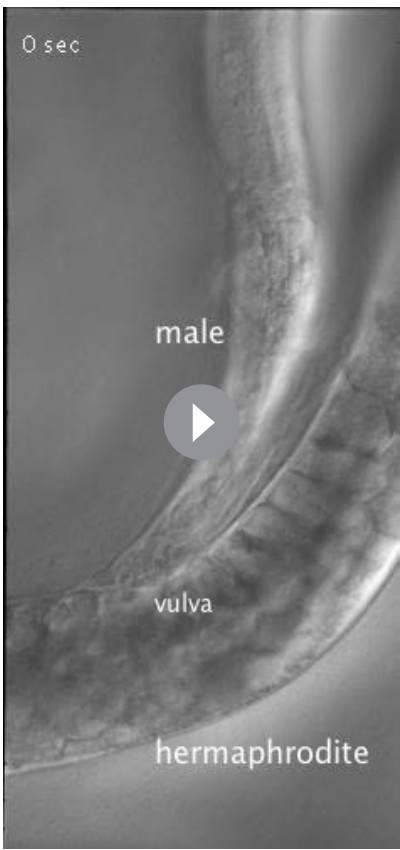

**Video 7.** Males performing Molina manoeuvres during mating: PHD-ablated male performing a defective, discontinuous Molina manoeuvre. Videos are played at 40 fps (sped up x2).

https://elifesciences.org/articles/48361#video7

partner becomes slightly erratic, often interrupted by a switch in direction (during scanning), or its initiation is delayed by a stop (during Molina manoeuvres). The qualitative difference in locomotion defects between these two steps of mating may result from differences in the state of the neural network before and after spicule insertion attempts (*Correa et al., 2015*; *Correa et al., 2012*; *Liu et al., 2011*). Sometimes while pausing, particularly during Molina manoeuvres but also during scanning, *wildtype* and PHD-ablated males prodded their spicules at locations other than the vulva (*Figure 7—figure supplement 2*). Ectopic prodding happened rarely but it occurred significantly more often in ablated than in *wildtype* males (*Figure 7—figure supplement 2A*). Because ectopic prodding occurred only during pauses and PHD-ablated males produced many more discontinuous manoeuvres than *wildtype* males, we interpret the increase in ectopic prodding as a secondary consequence of pausing rather than as the main defect of PHD-ablated males. Consistent with this interpretation, there was no statistically significant difference in the occurrence of ectopic prodding events between ablated and *wildtype* males when only discontinuous manoeuvres were considered (*Figure 7—figure supplement 2B*).

In addition to the aforementioned changes in neuronal activity, the highest level of PHD activation was observed during intromission, the penultimate step of the mating sequence, which precedes sperm transfer and involves full insertion of the spicules into the mate's vulva while sustaining backward locomotion. PHD activity increased two-fold upon spicule insertion and remained high for several seconds while the spicules were inserted (*Figure 7H*). PHDs are required for the efficiency of these two last steps of mating since PHD-ablated males that were observed to complete the mating sequence, including intromission and ejaculation, during a single mating with one individual mate, produced fewer cross-progeny than intact males in the same conditions (*Figure 7H*). Intact PHDs may increase the efficiency of sperm transfer by controlling the male's posture during intromission. While the cues and network interactions that activate PHDs during mating are presently not known, the high levels of PHD activity upon spicule insertion are consistent with a putative proprioceptive role for these neurons.

## Discussion

The data presented here extend John Sulston's original observations and demonstrate that male PHso1 cells undergo a direct glia-to-neuron transdifferentiation to produce a novel class of bilateral cholinergic, peptidergic, and ciliated sensory neurons, the phasmid D neurons. This updates the anatomy of the adult *C. elegans* male: it now comprises 387 neurons (93 of which are male-specific) and 90 glia (two are lost during the transdifferentiation to the PHDs). This is the second example of neurons arising from glia in *C. elegans*, confirming that glia can act as neural progenitors across metazoan taxa. In both cases we have demonstrated that fully differentiated, functional glia can retain neural progenitor properties during normal development. The complete glia-to-neuron cell fate switches we observe strongly suggest the production of neurons from glia is a process of natural transdifferentiation. In the first case (*Sammut et al., 2015*), this is an indirect transdifferentiation as it occurs via an asymmetric cell division, leading to self-renewal of the glial cell and the production of a neuron. In contrast, the work described here characterises a direct transdifferentiation and as such is the second direct transdifferentiation to be described in *C. elegans*. The first involves the transdifferentiation of the rectal epithelial cell Y into the motor neuron PDA, which occurs in hermaphrodites at earlier larval stages, before sexual maturation (*Jarriault et al., 2008*). Excitingly, most of the key genes that regulate the Y-to-PDA cell fate switch (*Kagias et al., 2012*; *Richard et al., 2011*; *Zuryn et al., 2014*) appear dispensable for PHso1-to-PHD and AMso-to-AMso +MCM transdifferentiation, indicating that independent molecular mechanisms exist that control cell-fate plasticity in a sex-specific, time-specific and/or a cell-specific manner. Intriguingly, direct conversion of glial-like neural stem cells has been observed in the adult zebrafish brain (*Barbosa et al., 2015*), suggesting that this may be a conserved mechanism for post-embryonic generation of neurons. Furthermore, it will be interesting to understand the molecular basis underlying the variability in the low frequency, background-dependent division of PHso1 and whether it has any functional or evolutionary consequence. The fact that a cell's terminal division can be variable is especially surprising in *C. elegans*, as its cell lineage is considered highly stereotyped and invariable. The only other neurons known to be variable in their generation are the post-embryonic and male-specific DX3/4 and EF3/4 interneurons located in the pre-anal ganglion (*Sulston et al., 1983*). It is intriguing that the EF neurons are the main post-synaptic partners of PHD (*Figure 6D*), and as such it is tempting to speculate there may be a possible link between cell division and cell connectivity.

Based on our manipulations and imaging of PHD activity in restrained and freely mating animals, we propose that PHDs may be proprioceptive neurons which become activated by tail-tip deformations and engage the circuits for backward locomotion. The high levels of PHD activity during intromission and upon tail-tip bending during the Molina manoeuvre, steps which require inhibition of forward locomotion and sustained backward movement, are consistent with such a model. PHDs may ensure sustained backward locomotion through excitation of their post-synaptic targets PVY, PVX, and EF interneurons, all of which have been shown to promote backward movement through AVA and AVB during scanning and location of vulva (*Sherlekar and Lints, 2014*; *Sherlekar et al., 2013*). As *wildtype* hermaphrodites are highly uncooperative during mating and actively move away (*Kleemann and Basolo, 2007*), any erratic and uncoordinated movement by the male is likely to result in the loss of its mate.

Sensory input from the mating partner is essential for successful mating and accordingly, many sensory neurons in copulation circuits are dedicated to this purpose (reviewed in *Villella and Hall, 2008* and *Barr et al., 2018*). However, coordinated motor control in all animals also requires proprioception, sensory feedback from internal tissues that inform the individual about its posture and strength exerted during movement (*Sherrington, 1907*; *Tuthill and Azim, 2018*). Several putative proprioceptive neurons have been identified in *C. elegans* (reviewed in *Schafer, 2015*). Within the male copulation circuit, a putative proprioceptive role has been attributed only to the spicule neuron SPC, on the basis of the attachment of its dendrite to the base of the spicules (*Sulston et al., 1980*). The SPC neuron is required for full spicule protraction during intromission (*García et al., 2001*; *Liu and Sternberg, 1995*). The presence of proprioceptive neurons in copulation circuits may be a broadly employed mechanism to regulate behavioural transitions during mating. In this light, it would be interesting to determine whether the mechanosensory neurons found in the mating circuits of several other organisms (*Liu et al., 2007*; *Ng and Kopp, 2008*; *Pavlou et al., 2016*) may play a role in self-sensory feedback.

In summary our results provide new insight into the developmental mechanisms underlying the neural substrates of sexually dimorphic behaviour. The glia-to-neuron transdifferentiation that results in PHD represents an extreme form of sexual dimorphism acquired by differentiated sex-shared glial cells during sexual maturation. The PHD neurons, perhaps through proprioception, enable the smooth and coordinated readjustment of the male's movement along its mate when spicule insertion becomes difficult to attain. This readjustment movement represents an alternative step within the stereotyped mating sequence. These findings reveal the extent to which both cell fate and innate behaviour can be plastic yet developmentally wired.

## Materials and methods

### Key resources table

| Reagent type (species) or resource | Designation | Source or reference | Identifiers | Additional information |
|---|---|---|---|---|
| Genetic reagent (*C. elegans*) | *C. elegans*: Strain N2 | Caenorhabditis Genetics Center | WormBase: N2 | |
| Genetic reagent (*C. elegans*) | AW827 | Dr Alison Woollard laboratory | | *him-5(e1490), heIs63 [wrt-2p::GFP::PH + wrt-2p::GFP::H2B + lin-48p::mCherry] V* |
| Genetic reagent (*C. elegans*) | CHL56 | This study | | *drpIs3 [lin-48p::tdTomato] I; him-5(e1490) V* |
| Genetic reagent (*C. elegans*) | BAR37 | This study | | *nsIs198 [mir-228p::GFP + lin-15(+)]; otIs356 [rab-3p::2xNLS::TagRFP], him-5(e1490) V* |
| Genetic reagent (*C. elegans*) | CHL32 | This study | | *dpy-5(e907) I; otIs356 [rab-3p::2xNLS::TagRFP], him-5 (e1490) V; drpIs1 [grl-2p::GFP, dpy-5(+)] X* |
| Genetic reagent (*C. elegans*) | OH13083 | Pereira et al. Elife. 2015; Caenorhabditis Genetics Center | | *him-5(e1490) V; otIs576 [unc-17fos::GFP + lin-44::YFP]* |
| Genetic reagent (*C. elegans*) | CHL36 | This study | | *drpIs3 [lin-48p::tdTomato] I; inIs179 [ida-1p::GFP] II; him-8(e1489) IV* |
| genetic reagent (*C. elegans*) | CHL57 | This study | | *drpIs3 [lin-48::tdTomato] I; mnIs17 [osm-6p::GFP + unc-36(+)]; him-5(e1490) V* |
| Genetic reagent (*C. elegans*) | CHL58 | This study | | *drpIs3 [lin-48::tdTomato] I; otIs291 [rab-3p::2xNLS::YFP + rol-6(su1006)], him-5(e1490) V* |
| Genetic reagent (*C. elegans*) | CHL59 | This study | | *drpIs3 [lin-48p::tdTomato] I; dpy-17(e164) III; otIs291 [rab-3p::2xNLS::YFP + rol-6(su1006)], him-5(e1490) V* |
| Genetic reagent (*C. elegans*) | CHL60 | This study | | *drpIs3 [lin-48p::tdTomato] I; ced-4(n1162), dpy-17(e164) III; otIs291 [rab-3p::2xNLS::YFP + rol-6(su1006)], him-5(e1490) V* |
| Genetic reagent (*C. elegans*) | CHL61 | This study | | *drpIs3 [lin-48p::tdTomato] I; him-5(e1490) V; drpIs1 [grl-2p::GFP, dpy-5(+)] X* |
| Genetic reagent (*C. elegans*) | CHL63 | This study | | *drpIs3 [lin-48p::tdTomato] I; him-5(e1490) V; vsIs48 [unc-17p::GFP] X* |
| Genetic reagent (*C. elegans*) | CHL64 | This study | | *drpIs3 [lin-48p::tdTomato] I; inIs179[ida-1p::GFP] II; him-5(e1490) V* |

*Continued on next page*

*Continued*

| Reagent type (species) or resource | Designation | Source or reference | Identifiers | Additional information |
|---|---|---|---|---|
| Genetic reagent (*C. elegans*) | CHL67 | This study | | *drpIs3 [lin-48p::tdTomato] I; drpIs4 [oig-8p::GFP + pha-1(+)]; him-5(e1490) V* |
| Genetic reagent (*C. elegans*) | CHL65 | This study | | *drpIs4 [oig-8p::GFP + pha-1(+)]; him-5(e1490) V* |
| Genetic reagent (*C. elegans*) | BAR77 | This study | | *oleEx24 [grl-2(1 kb)::fem-3::SL2::mCherry (8 ng)+ elt-2p::GFP (40 ng)]; inIs179 [ida-1p::GFP] II; him-8(e1489) IV* |
| Genetic reagent (*C. elegans*) | CHL62 | This study | | *oleEx18 [grl-2(1 kb)::fem-3::SL2:: mCherry(20 ng) + elt-2p::GFP (40 ng)]; drpIs3 [lin-48p::tdTomato] I; otIs291 [rab-3p::2xNLS::YFP + rol-6(su1006)], him-5(e1490) V* |
| Genetic reagent (*C. elegans*) | CHL68 | This study | | *sem-4(n19719), drpIs3 [lin-48p::tdTomato] I; otIs291 [rab-3p::2xNLS::YFP + rol-6(su1006)], him-5(e1490) V* |
| Genetic reagent (*C. elegans*) | CHL69 | This study | | *drpIs3 [lin-48p::tdTomato] I; egl-27(ok1670) II; otIs291 [rab-3p::2xNLS::YFP + rol-6(su1006)], him-5(e1490) V* |
| Genetic reagent (*C. elegans*) | CHL70 | This study | | *drpIs3 [lin-48p::tdTomato] I; otIs291 [rab-3p::2xNLS::YFP + rol-6(su1006)], him-5(e1490) V; nre-1(hd20), lin-15B(hd126) X* |
| Genetic reagent (*C. elegans*) | CHL71 | This study | | *drpIs3 [lin-48p::tdTomato] I; inIs179 [ida-1p::GFP] II; him-8(e1489) IV* Generated during *sox-2* mutant crosses. Distinct background to CHL36. |
| Genetic reagent (*C. elegans*) | CHL72 | This study | | *drpIs3 [lin-48p::tdTomato] I; inIs179 [ida-1p::GFP] II; him-8(e1489) IV; otEx4454 [sox-2(fosmid)::mCherry + elt-2p::DsRed]* |
| Genetic reagent (*C. elegans*) | CHL73 | This study | | *drpIs3 [lin-48p::tdTomato] I; inIs179 [ida-1p::GFP] II; him-8(e1489) IV; sox-2(ot640[unc-119(+)]) X; otEx4454 [sox-2(fosmid)::mCherry + elt-2p::DsRed]* |
| Genetic reagent (*C. elegans*) | CHL74 | This study | | *drpIs3 [lin-48p::tdTomato] I; inIs179 [ida-1p::GFP] II; him-8(e1489) IV; nre-1(hd20), lin-15B(hd126) X* |
| Genetic reagent (*C. elegans*) | EM1370 | Dr Scott Emmons laboratory | | *bxEx271 [oig-8p::GFP + oig-8p::mCherry::rab-3]; him-5(e1490) V* |
| Genetic reagent (*C. elegans*) | EM1371 | Dr Scott Emmons laboratory | | *bxEx272 [oig-8p::GFP + oig-8p::mCherry::rab-3]; him-5(e1490) V* |
| Genetic reagent (*C. elegans*) | BAR90 | This study | | *him-5(e1490) V; oleEx27 [oig-8p::GCaMP6f::SL2::mRFP(30 ng/μl) + cc::GFP(30 ng/μl)]* |
| Genetic reagent (*C. elegans*) | BAR115 | This study | | *unc-51(e369), him-5(e1490) V; oleEx27 [oig-8p::GCaMP6f:: SL2::mRFP(30 ng/μl) + cc::GFP(30 ng/μl)]* |

*Continued on next page*

*Continued*

| Reagent type (species) or resource | Designation | Source or reference | Identifiers | Additional information |
|---|---|---|---|---|
| Genetic reagent (*C. elegans*) | BAR115 | This study | | *unc-51(e369), him-5(e1490) V; oleEx27 [oig-8p::GCaMP6f:: SL2::mRFP(30 ng/µl) + cc::GFP(30 ng/µl)]; oleEx38 [myo-3::HisCL1::SL2::Cherry(20 ng/µl)]* |
| Genetic reagent (*C. elegans*) | BAR95 | This study | | *unc-13 (e51) I; him-5 (e1490) V; oleEx27 [oig-8p::GCaMP6f:: SL2::mRFP(30 ng/µl) + cc::GFP(30 ng/µl)]* |
| Genetic reagent (*C. elegans*) | BAR106 | This study | | *unc-31(e169) IV; him-5(e1490) V; oleEx27 [oig-8p::GCaMP6f:: SL2::mRFP(30 ng/µl) + cc::GFP(30 ng/µl)]* |
| Genetic reagent (*C. elegans*) | EM1251 | Dr Scott W. Emmons laboratory | | *bxEx201[oig-8p::GFP + pha-1(+)]; him-5(e1490) V - line#1* |
| Genetic reagent (*C. elegans*) | EM1253 | Dr Scott W. Emmons laboratory | | *bxEx201[oig-8p::GFP + pha-1(+)]; him-5(e1490) V - line#3* |
| Genetic reagent (*C. elegans*) | BAR94 | This study | | *him-5(e1490) V; lite-1(ce314) X; oleEx30[oig-8p::GCaMP6f::SL2::mRFP(50 ng/µl)]* |
| Genetic reagent (*C. elegans*) | CB369 | Caenorhabditis Genetics Center | | *unc-51(e369) V* |
| Genetic reagent (*C. elegans*) | BAR160 | This study | | *him-5(e1490) V; oleEx53 [pAB6(oig-8p::HisCl1::SL2::RFP) (30 ng/uL)+unc-122p::GFP(25 ng/uL)]* |
| Strain, strain background (*E. coli*) | Strain OP50 | Caenorhabditis Genetics Center | OP50 | |
| Strain, strain background (*E. coli*) | Strain MG1693 (Thy-) | *E. coli* stock centre | MG1693 | Strain for EdU staining experiments |
| Strain, strain background (*E. coli*) | Strain: HT115 (DE3) | Caenorhabditis Genetics Center | HT115 | |
| Tecombinant DNA reagent | Plasmid: pPD95.75 | Addgene | #1494 | See: *DNA constructs and transgenic strains* Used to create *oig-8prom::GFP* and for PCR fusions |
| Recombinant DNA reagent | Plasmid: *oig-8prom::gfp* | This study | | See: *DNA constructs and transgenic strains* 964 bp upstream of the *oig-8* ATG |
| Recombinant DNA reagent | Plasmid: *pkd-2prom::mCherry::rab-3* | Dr María I. Lázaro-Peña | | See: *DNA constructs and transgenic strains* Replaced promoter by *oig-8prom* |
| Recombinant DNA reagent | Plasmid: *oig-8prom::mCherry::rab-3* | This study | | See: *DNA constructs and transgenic strains* Created from *pkd-2prom::mCherry::rab-3* |
| Recombinant DNA reagent | Plasmid pLR306 | Dr Luis Rene García Laboratory | | See: *DNA constructs and transgenic strains* Gateway plasmid used to amplify *GCaMP6f::sl2::rfp* |
| Recombinant DNA reagent | Plasmid: *oig-8prom::GCaMP6f::SL2::RFP* | This study | | See: *DNA constructs and transgenic strains* Created by PCR fusion (see primers) |
| Recombinant DNA reagent | Plasmid: pNP471 | Dr Cori Bargmann Laboratory | | See: *DNA constructs and transgenic strains* Used to amplify *HisCl1::sl2::mCherry* |
| Recombinant DNA reagent | Plasmid: *oig-8prom::HisCl1::sl2::rfp* | This study | | See: *DNA constructs and transgenic strains* Created by Gibson |
| Recombinant DNA reagent | Plasmid: *myo-3prom::HisCl1::sl2::mCherry* | This study | | See: *DNA constructs and transgenic strains* Created by PCR fusion (see primers) |
| Recombinant DNA reagent | Plasmid: *lin-48prom::tdTomato* | Dr Mike Boxem Laboratory | | See: *DNA constructs and transgenic strains* 6.8 kb upstream of *lin-48* ATG |
| Recombinant DNA reagent | Plasmid pPD129.36 (L4440) | Dr Andrew Fire Laboratory, Addgene | #1654 | Control plasmid for RNAi experiments |

*Continued on next page*

*Continued*

| Reagent type (species) or resource | Designation | Source or reference | Identifiers | Additional information |
|---|---|---|---|---|
| Recombinant DNA reagent | Plasmid: *sox-2* RNAi clone | Source BioScience | | Silence endogenous *sox-2* gene |
| Sequence-based reagent | primer_oig-8 fusion_F | This study | PCR fusion primers | See: *DNA constructs and transgenic strains* gggagtgacctatgcaaacc |
| Sequence-based reagent | primer_oig-8 fusion_R | This study | PCR fusion primers | See: *DNA constructs and transgenic strains* CGACGTGATGAGTCGACCAT tgttttacctgaaatctttt |
| Sequence-based reagent | primer_myo-3 fusion_F | This study | PCR fusion primers | See: *DNA constructs and transgenic strains* cgtgccatagttttacattcc |
| Sequence-based reagent | primer_myo-3 fusion_R | This study | PCR fusion primers | See: *DNA constructs and transgenic strains* gctagttgggctttgcatGCttctagatggatctagtggtc |
| Sequence-based reagent | primer_ced-4 (n1162)_F | This study | PCR genotyping primers | See: *C. elegans strains* gatgctctgcgaaatcgaatgc |
| Sequence-based reagent | primer_ced-4 (n1162)_R | This study | PCR genotyping primers | See: *C. elegans strains* tcacctaaatcacacatctcgtcg |
| Sequence-based reagent | primer_dpy-17 (e164)_F | This study | PCR genotyping primers | See: *C. elegans strains* aggaggaagcccaatcaacc |
| Sequence-based reagent | primer_dpy-17 (e164)_R | This study | PCR genotyping primers | See: *C. elegans strains* cagttggtccttcttctccagc |
| Sequence-based reagent | primer_sem-4 (n1971)_F | This study | PCR genotyping primers | See: *C. elegans strains* aaggtgatgcgatgatgtctcc |
| Sequence-based reagent | primer_sem-4 (n1971)_R | This study | PCR genotyping primers | See: *C. elegans strains* taatgatcggcttgggtgtgg |
| Sequence-based reagent | primer_egl-27 (ok1679)_F ext | This study | PCR genotyping primers | See: *C. elegans strains* tcatcgtttccagtctcttcagg |
| Sequence-based reagent | primer_egl-27 (ok1679)_R ext | This study | PCR genotyping primers | See: *C. elegans strains* cgctggttatcaaatgacgcc |
| Sequence-based reagent | primer_egl-27 (ok1679)_F int | This study | PCR genotyping primers | See: *C. elegans strains* agacaccagaagctacgaaacc |
| Sequence-based reagent | primer_egl-27 (ok1679)_R int | This study | PCR genotyping primers | See: *C. elegans strains* gtttgcatcacggtcttcacg |
| Sequence-based reagent | primer_nre-1(hd20) lin-15B(hd126)_F | This study | PCR genotyping primers | See: *C. elegans strains* catgagagctgcgctgaggc |
| Sequence-based reagent | primer_nre-1(hd20) lin-15B(hd126)_R | This study | PCR genotyping primers | See: *C. elegans strains* ggttcgggctcgcggtagtc |
| Chemical compound, drug | Paraformaldehyde | Thermo Fisher Scientific | #28908 | Used at 4% |
| Chemical compound, drug | β-mercapto-ethanol | Sigma Aldrich | M6250 | Used at 5% |
| Chemical compound, drug | Collagenase type IV | Sigma Aldrich | C-5138 | Used at 1 mg/ml |
| Chemical compound, drug | Vectashield antifade | Vector Laboratories | H-1900 | |
| Chemical compound, drug | Isopropyl-β-D-thiogalactopyranoside (IPTG) | Generon | GEN-S-02122 | Used at 0.6 mM in plates for RNAi knock-down |
| Chemical compound, drug | Histamine | Sigma | H7125 | Used at 20 mM in plates for behavioural assays |
| Chemical compound, drug | Wormglu | GluStitch, Delta, Canada | | |
| Antibody | Anti-RFP pAb (Rabbit polyclonal antibody) | MBL International | PM005 | (1:500) |

*Continued on next page*

*Continued*

| Reagent type (species) or resource | Designation | Source or reference | Identifiers | Additional information |
|---|---|---|---|---|
| Antibody | Donkey anti-Rabbit IgG (H+L) Highly Cross-Adsorbed Secondary Antibody, Alexa Fluor 555 (Donkey polyclonal antibody) | Molecular probes | A-31572 | (1:200) |
| Commercial assay or kit | Click-IT EdU Alexa Fluor 594 | Invitrogen | C10339 | |
| Software, algorithm | ImageJ | ImageJ (http://imagej.nih.gov/ij/) | RRID:SCR_003070 | |
| Software, algorithm | GraphPad Prism | GraphPad Prism (graphpad.com) | RRID:SCR_002798 | |
| Software, algorithm | Affinity Designer | https://affinity.serif.com/en-us/designer/ | RRID:SCR_016952 | |

## *C. elegans* strains

Nematode culture and genetics were performed following standard procedures (*Brenner, 1974*; *Stiernagle, 2006*). For genetic crosses, all genotypes were confirmed using PCR unless the mutant phenotype was obvious (see Key Resource Table for primer sequences). Animals considered *wild-type* carry *him-5* or *him-8* mutations to facilitate male analysis. All assays were conducted at 20°C. Strains used in this study are listed in the Key Resource Table.

## DNA constructs and transgenic strains

The *oig-8^prom^::gfp* reporter was built by amplifying a 964 bp promoter fragment upstream of the *oig-8* ATG adding the restriction sites of SphI and XmaI (see Key Resource Table for these and other primer sequences). This promoter drives expression in PHD neurons and two pairs of sensory neurons in the head which we have not identified. The PCR product was digested and ligated into SphI/XmaI-digested pPD95.75 vector. To generate *oig-8^prom^::mCherry::rab-3*, the same *oig-8* promoter fragment was ligated into SphI/XmaI-digested *pkd-2^prom^::mCherry::rab-3* (a gift from M. Lázaro-Peña). The *oig-8^prom^::GCaMP6f::SL2::rfp* construct was created by PCR fusion (*Hobert, 2002*). A ~ 1 kb promoter region of the *oig-8* locus was PCR fused with the GCaMP sequence. The *GCaMP6f::sl2::rfp* fragment was amplified from gateway plasmid pLR306, a kind gift from Luis Rene García. The same promoter region was used to create the *oig-8^prom^::HisCl1::sl2::rfp* plasmid by Gibson cloning. The *myo-3^prom^::HisCl1::sl2::mCherry* construct was created as PCR fusion. A 2.2 kb promoter region of the *myo-3* locus was amplified and PCR fused with the *HisCl1* sequence (see Key Resource Table for primer sequences). The *HisCl1::sl2::mCherry* fragment was amplified from plasmid pNP471, a kind gift from the Bargmann lab. The *grl-2^prom^::gfp* reporter was generated by integrating sEx12852 (*Hao et al., 2006*) to generate *drpIs1*. The *lin-48^prom^::tdTomato* construct was a kind gift from Mike Boxem and contains a 6.8 kb promoter fragment upstream of the coding sequence of *tdTomato* and the *unc-54* 3' UTR. It was injected into *him-5(e1490)* animals and spontaneously integrated generating *drpIs3*.

## PHso1-to-PHD single animal live imaging

Animals containing the *lin-48^prom^::tdTomato drpIs3* transgene were synchronised via sodium hypochlorite treatment and allowed to develop until the early L3 stage. Next, they were mounted individually on 5% agarose pads in a droplet of the minimum concentration of sodium azide that paralysed the animals (which varied within experimental replicates) and viewed under a compound microscope to determine their developmental time. A total of 10–20 males in which the gonad had just turned to grow posteriorly were imaged. Care was taken to use the lowest fluorescent light intensity and exposure in order to minimise toxicity. After the initial imaging, animals were gently recovered by aspiration, washed in a droplet of M9 buffer, transferred to individual plates and allowed to develop overnight at 15°C. Each animal was then immobilised and imaged as described above. In those animals where the leading edge of the gonad had crossed back over to the ventral side, a sequence of

time-lapse images was taken every 2–4 hr until PHso1-to-PHD remodelling was complete, 10–12 hr later.

## Immunohistochemistry and EdU staining

EdU (5-ethynyl-2′-deoxyuridine) feeding was performed as previously described (*van den Heuvel and Kipreos, 2012*) combined with immunohistochemistry to enhance reporter transgene fluorescence. In short, MG1693 bacteria (Thy-deficient, *E. coli* stock centre: http://cgsc.biology.yale.edu/) were grown in 100 ml of minimal medium containing 20 mM EdU for at least 24 hr, concentrated by centrifugation and seeded on NGM plates with 100 µg/mL ampicillin. A synchronised population of L1 larvae, obtained by hypochlorite treatment and hatching in M9 buffer, was allowed to feed on OP50 *E. coli* until early L3 (24–28 hr at 20°C). Worms were washed thoroughly in M9, plated onto MG1693 EdU containing plates and allowed to grow until late L4 – early adult stage (22–24 hr). Hard-tube fixation was chosen in order to preserve the male tail morphology (*McIntire et al., 1992*). Worms were washed thoroughly, transferred to DNA LoBind tubes (Eppendorf) and fixed for 12 hr in 4% paraformaldehyde at 4°C in a rocking nutator, followed by 5% β-mercapto-ethanol for another 12 hr at 37°C in a rocking nutator. Worms were then incubated for 6 hr in 1 mg/ml collagenase (Sigma Aldrich) at 37°C and 700 rpm, followed by 24 hr with rabbit anti-RFP antibody (1:500, MBL) to detect the tdTomato fluorescence from the *drpIs3* reporter transgene. Alexa 555-conjugated donkey anti-rabbit (1:200; Molecular probes) was used as secondary antibody. As EdU reaction can interfere with antibody staining, this was performed after the immunohistochemistry. The Click-IT reaction was performed in a total volume of 50 µl per tube, in the dark, according to the manufacturer's instructions (Click-IT EdU Alexa Fluor 594 kit, Invitrogen). Slides were mounted in Vectashield antifade mounting medium (Vector Laboratories). Samples were analysed the same day or the day after mounting, to minimise fading of fluorescence. Only animals where both head and tail were properly stained for tdTomato (cell body) and EdU (nucleus) were scored. Four independent EdU experiments were performed (see Source Data *Figure 3*) and all animals scored were plotted in the same bar chart.

## RNAi feeding experiments

The RNAi bacterial clone for *sox-2* was obtained from the Ahringer RNAi Collection (Source BioScience) (*Kamath et al., 2003*). RNAi feeding experiments were performed following standard protocols (*Kamath et al., 2001*). Briefly, isopropyl β-D-1-thiogalactopyranoside (IPTG) was added to a final concentration of 0.6 mM to NGM medium to prepare the RNAi plates. 24 hr later, HT115 bacteria transfected with the *sox-2* RNAi clone was seeded onto RNAi plates. Staged early-L4 larvae were transferred to seeded RNAi plates and 4–5 days later their progeny, which had developed under the embryonic effects of RNAi knock-down (F1 generation), were scored at the stage of young adult. To exclusively analyse post-embryonic effects, a synchronised population of L1 larvae was transferred to RNAi seeded plates and young adult males were scored 2–3 days later (P0 generation).

All experiments were performed at 20°C. The *nre-1(hd20) lin-15b(hd126)* background was used to sensitise worms to the RNAi effects. One experiment was performed with three replicate plates per condition. L4440 empty vector (pPD129.36, Addgene) was used as negative control and *gfp* RNAi clone as positive control.

## Cell-type specific sex transformations

We used two previously described strains each containing an array that drives *fem-3* expression from a *grl-2* promoter fragment (*Sammut et al., 2015*). This *grl-2* promoter is expressed in PHso1 and PHso2 in the tail, and AMso, excretory pore, and excretory duct in the head (*Hao et al., 2006*). In the case of *oleEx24* the presence/absence of the array in PHso1 and PHso2 was monitored by the *mCherry* expression from the array itself and the expression of an *ida-1^prom^::gfp* reporter (*inIs179*), in the background of the strain was used to monitor neuronal fate in PHso1/PHD. In the case of *oleEx18*, we had difficulty visualising the *mCherry* from the array in PHD and crossed a *lin-48^prom^:: tdTomato* transgene (*drpIs3*) into the strain to visualise PHso1/PHD. The presence/absence of the array in whole animals was assessed using the array co-injection marker *elt-2^prom^::gfp*. Neuronal fate was monitored using a *rab-3^prom^::yfp* reporter.

## Cell ablations

PHD was ablated with a laser microbeam as previously described (*Bargmann and Avery, 1995*). L4 males were staged and placed on a seeded plate the night before. Ablations were carried out at one-day of adulthood and PHD was identified by *oig-*$^{prom}$*::gfp* or *unc-17*$^{prom}$*::gfp* reporter expression. Mock-ablated animals underwent the same treatment as ablated males except for laser trigger. Animals were left to recover overnight and behavioural assays were performed the next day. After behavioural assays, animals were checked for lack of GFP expression in the tail-tip region where PHD is normally located to confirm correct cell ablation. The few animals in which PHD had not been efficiently ablated (judged by the presence/return of GFP) were discarded from the data.

## Behavioural assays

All behavioural assays were scored blind to the manipulation. Males carried either an *oig-8*$^{prom}$*::gfp* or an *unc-17*$^{prom}$*::gfp* transgene to identify the PHD neurons for ablation. For the experiments in which PHD was silenced with a HisCl1 transgene, mating assays were performed in 2-day-old adults, as in the ablations. Before the assay, *wildtype* or array-carrying animals with RFP-positive PHD neurons were moved to a 20 mM histamine plate with food for 1 hr. As controls, array-carrying animals not treated with histamine were also tested. Histamine plates were prepared as regular NGM plates adding 20 mM histamine when the agar cooled to a temperature below 56°C.

### Food leaving

Animals were tested as 3-day-old adults (the day after they had been tested for mating). Assays were performed and scored as previously described (*Barrios et al., 2008*).

### Mating

Assays were performed and scored as previously described (*Sammut et al., 2015*). Males were tested at two days of adulthood with 1-day-old *unc-51(e369)* hermaphrodites picked the night before as L4s. Each male was tested once for all steps of mating. Those males that were not successful at inserting their spicules with the first hermaphrodite were tested again with a maximum of three hermaphrodites to control for hermaphrodite-specific difficulty in penetration (*Liu and Sternberg, 1995*). Assays were replicated at least three times on different days and with different sets of males. Videos of mating events were recorded and scored blind by two independent observers.

### Response

A male was scored as responding to mate contact if it placed its tail ventral down on the mate's body and initiated the mating sequence by backing along the mate's body to make a turn. The response efficiency was calculated by dividing 1 (response) by the total number of contacts made with the mate before responding. If a male did not respond within three minutes, it was scored as having 0 efficiency. As a more sensitive measure of the quality of response, we scored hesitation during response. Hesitation is a switch in direction between forward and backward locomotion from the time the male establishes contact with the mate to the first turn (or to location of vulva if this occurs without the need of a turn).

### Scanning

A single scan was scored as the journey around the mate's body away from and returning to the vulva position. The first scan was counted as the journey from the point of first contact to the hermaphrodite vulva position. A scan was considered continuous if locomotion was maintained in the backward direction without switching direction or pausing.

### Ectopic prodding

Prodding was scored as a visible protrusion of the spicules out of the gubernaculum and/or visible twitching of the spicule muscles. Prodding was considered ectopic if it occurred at regions of the mate's body other than the vulva.

### Molina manoeuvres

A continuous single manoeuvre was scored as the journey away from the vulva in forward locomotion, to a distance bigger than two tail-tip lengths, and return to the vulva in backward locomotion. Since the transition between the forward and backward steps could take up to 3 s in a *wildtype* male, the category of discontinuous manoeuvre 'stop before backward step' was scored if the pause was longer than 3 s. Any other visible pause during forward or backward locomotion was considered a STOP regardless of its duration. The category of discontinuous manoeuvre 'switching' was scored as a change in direction of locomotion while travelling away or towards the vulva without reaching it.

### Displacements from the vulva

Scored as movements of one or two tail-tip lengths away from the vulva.

### Fertility assays

Each individual male was monitored for all mating steps during a single mating until it ejaculated. After disengagement from the mate, the hermaphrodite was picked and placed in a fresh plate to lay progeny. The adult hermaphrodite was transferred to a fresh plate each day during three consecutive days. After three days from the eggs being laid, L4 larvae and adult progeny were counted as Unc self-progeny of Wt cross-progeny.

## Microscopy and imaging

Worms were anesthetised using 50 mM sodium azide and mounted on 5% agarose pads on glass slides. Images were acquired on a Zeiss AxioImager using a Zeiss Colibri LED fluorescent light source and custom TimeToLive multichannel recording software (Caenotec). Representative images are shown following maximum intensity projections of 2–10 1 μm z-stack slices and was perfumed in ImageJ.

## Ca$^{2+}$ imaging

Imaging was performed in an upright Zeiss Axio Imager two microscope with a 470 nm LED and a GYR LED (CoolLED) with a dual-band excitation filter F59-019 and dichroic F58-019 (Chroma) in the microscope turret. Emission filters ET515/30M and ET641/75 and dichroic T565lprx-UF2 were placed in the cube of a Cairn OptoSplit II attached between the microscope and an ORCA-Flash four camera (Hamamatsu). Acquisition was performed at 20 fps. Imaging during mating was performed with a 20x long working distance objective (LD Plan-NEOFLUAR numerical aperture 0.4), placing the male on an agar pad with food and 20 hermaphrodites. The ~50 mm per side agar pad was cut out from a regular, seeded NGM plate and placed on a glass slide. The hermaphrodites were placed in a ~ 100 mm$^2$ centre region. A fresh pad was used every two recordings.

Imaging in restrained animals was performed for 2.5–3 min with a 63x objective (LD C- apochromat numerical aperture 1.15). Animals were glued with Wormglu along the body to a 5% agarose pad on a glass slide and covered with M9 or 20 mM histamine (Sigma, H7125) and a coverslip.

## Ca$^{2+}$ imaging analysis

A moving region of interest in both channels was identified and mean fluorescent ratios (GFP/RFP) were calculated with custom-made Matlab scripts (*Busch et al., 2012*), kindly shared by Zoltan Soltesz and Mario de Bono. For recordings in restrained animals, bleach correction was applied to those traces in which an exponential decay curve fitted with an R square >0.6. Ratios for each recording were smoothened using an 8-frame rolling average. For $\Delta R/R_0$ values, $R_0$ for each recording period was calculated as the mean of the lowest 10$^{th}$ percentile of ratio values. Traces that were locked to behavioural transitions had their $\Delta R/R_0$ values added or subtracted such that all events had the same value at t = 0.

Peaks were identified manually by an observer blind to the genotype and treatment. Peaks were called as signals above 0.2 $\Delta R/R_{max}$ (where $R_{max}$ was calculated as the highest 5$^{th}$ percentile of ratio values), above 2σ from local basal and a minimum duration of 5 s.

## Electron microscopy and serial reconstruction

The samples were fixed by chemical fixation or high-pressure freezing and freeze substitution as previously described (*Hall et al., 2012*). Several archival print series from *wildtype* male tails in the MRC/LMB collection were compared to *wildtype* adult males prepared in the Hall lab, showing the same features overall. Ultrathin sections were cut using an RMC Powertome XL, collected onto grids, and imaged using a Philips CM10 TEM. The PHD cell bodies were identified in the EM sections based on position and morphology. This was followed by serial tracing of the projections to establish their morphology and connectivity. The method of quantitative reconstruction using our custom software is described in detail in *Jarrell et al., 2012* and *Xu et al., 2013*. The connectivity of the PHD neurons was determined from the legacy N2Y EM series (*Sulston et al., 1980*). Circuit diagrams were generated using Cytoscape (*Smoot et al., 2011*). PHD neuronal maps and connectivity tools are available at www.wormwiring.org.

## Acknowledgements

We thank Mario de Bono and Zoltan Soltesz for kindly sharing custom-made software for ratiometric analysis of $Ca^{2+}$ imaging in moving animals; Rene García for worm cartoons used in *Figure 7*; Ken Nguyen for help with EM; John G White and Jonathan Hodgkin for their help in transferring archival TEM data from the MRC/LMB to the Hall Lab at the Albert Einstein College of Medicine for long-term curation and study; Shai Shaham for sharing the *mir-228*$^{prom}$*::gfp* strain prior to publication; Mike Boxem, María Lázaro-Peña, Alison Woollard and Cori Bargmann for additional strains and reagents; Sheila Poole for edits on the manuscript; Baris Kuru for aid with behavioural experiments. Additional strains were obtained from the CGC, which is supported NIH grant P40 OD010440. Christopher Brittin was influential in designing and creating www.wormwiring.org. This work was supported by a Newton Fellowship from the Royal Society to LMG (NF160914), a Wellcome Trust PhD studentship to RB, Marie Curie CIG grant 618779 and Wellcome Trust Enhancement Funding (095722/Z/11/A) to RJP, Leverhulme Trust grant RPG-2018–287 to AB, NIH R01 GM066897 grant and the G Harold and Leila Y Mathers Charitable Foundation to SWE, NIH OD 010943 to DHH, NIH T32GM007491 and F32 MH115438 01 to SJC; RJP was supported by a Wellcome Trust Research Career Development Fellowship (095722/Z/11/Z) and is currently a Wellcome Senior Fellow in Basic Biomedical Science (207483/Z/17/Z). AB and RJP are members of COST Action BM1408.

## Additional information

### Funding

| Funder | Grant reference number | Author |
|---|---|---|
| Royal Society | Newton Fellowship NF160914 | Laura Molina-Garcia |
| Wellcome Trust | PhD Studentship | Rachel C Bonnington |
| National Institutes of Health | T32 GM007491 | Steven J Cook |
| National Institutes of Health | F32 MH115438 | Steven J Cook |
| National Institutes of Health | OD 010943 | David H Hall |
| National Institutes of Health | R01 GM066897 | Scott W Emmons |
| Mathers Foundation | | Scott W Emmons |
| Wellcome Trust | Research Career Development Fellowship 095722/Z/11/Z | Richard J Poole |
| Wellcome Trust | Enhancement Funding 095722/Z/11/A | Richard J Poole |
| Wellcome Trust | Senior Research Fellowship 207483/Z/17/Z | Richard J Poole |

The funders had no role in study design, data collection and interpretation, or the decision to submit the work for publication.

## Author contributions

Laura Molina-García, Conceptualization, Resources, Formal analysis, Supervision, Funding acquisition, Investigation, Visualization, Writing - review and editing, LMG performed, analysed and interpreted the behavioural experiments; Carla Lloret-Fernández, Conceptualization, Resources, Formal analysis, Supervision, Investigation, Visualization, Writing - review and editing, CLF characterised the direct PHso1-to-PHD transdifferentiation, analysed mutant alleles and performed the Edu experiments; Steven J Cook, Resources, Formal analysis, Supervision, Funding acquisition, Investigation, Visualization, SJC with SWE reconstructed the connectivity; Byunghyuk Kim, Resources, Formal analysis, Funding acquisition, Investigation, Visualization, BK also identified neuronal characteristics in PHD and generated the oig-8 reporter constructs; Rachel C Bonnington, Formal analysis, Supervision, Funding acquisition, Investigation, Visualization, Writing - review and editing, RCB performed the sex-transformation experiments and with MS and with JMO performed the characterisation of the PHso1-to-PHD molecular transdifferentiation; Michele Sammut, Resources, Investigation, Visualization, Writing - review and editing, MS performed single-animal imaging experiments; Jack M O'Shea, Investigation, Visualization; Sophie PR Gilbert, Formal analysis, Investigation, Visualization, SPRG helped characterise lin-48 expression; David J Elliott, Investigation, DJE provided technical assistance; David H Hall, Resources, Data curation, Supervision, Funding acquisition, Investigation, Visualization, DHH provided and analysed the electron micrographs; Scott W Emmons, Resources, Supervision, Funding acquisition, Investigation; Arantza Barrios, Conceptualization, Resources, Formal analysis, Supervision, Funding acquisition, Investigation, Visualization, Writing - original draft, Project administration, Writing - review and editing, AB conceived, performed and interpreted the behavioural and Ca2+ imaging experiments and co-wrote the manuscript; Richard J Poole, Conceptualization, Resources, Formal analysis, Supervision, Funding acquisition, Investigation, Visualization, Writing - original draft, Project administration, Writing - review and editing, RJP conceived and performed the analysis of the PHso1-to-PHD transdifferentiation and co-wrote the manuscript

## Author ORCIDs

Laura Molina-García (ID) https://orcid.org/0000-0002-1530-4061
Carla Lloret-Fernández (ID) https://orcid.org/0000-0002-5370-9318
Steven J Cook (ID) https://orcid.org/0000-0002-1345-7566
Byunghyuk Kim (ID) https://orcid.org/0000-0002-2220-4173
Rachel C Bonnington (ID) https://orcid.org/0000-0002-5652-0297
Michele Sammut (ID) https://orcid.org/0000-0002-0281-6121
Jack M O'Shea (ID) https://orcid.org/0000-0002-9694-7340
David H Hall (ID) https://orcid.org/0000-0001-8459-9820
Scott W Emmons (ID) https://orcid.org/0000-0002-6511-9972
Arantza Barrios (ID) https://orcid.org/0000-0001-6153-8830
Richard J Poole (ID) https://orcid.org/0000-0001-6414-2479

## Decision letter and Author response

Decision letter https://doi.org/10.7554/eLife.48361.sa1
Author response https://doi.org/10.7554/eLife.48361.sa2

## Additional files

### Supplementary files

• Supplementary file 1. PHD connectivity. A table indicating the quantification of pre-synaptic inputs and post-synaptic outputs of the PHD neurons as derived from the electron microscopy serial reconstruction (see Materials and methods). The number of synapses and number of sections over which synapses are observed for each input/output neuron are indicated.

• Transparent reporting form

## Data availability

All data generated or analysed during this study are included in the manuscript and supporting files.

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
