## [Decision Letter]

**Acceptance summary:**

In this work, the authors add to findings showing that neurons can arise via transdifferentiation from other cell types in *C. elegans*. Here, they show that a sex-shared socket glial cell transdifferentiates to a male-specific sensory neuron type, and suggest that these neurons play a proprioceptive role in male mating behavior. This work provides insights into mechanisms by which the nervous system generates sexually dimorphic behaviors.

**Decision letter after peer review:**

Thank you for submitting your article "A direct glia-to-neuron natural transdifferentiation ensures nimble sensory-motor coordination of male mating behaviour" for consideration by *eLife*. Your article has been reviewed by three peer reviewers, and the evaluation has been overseen by a Reviewing Editor and Ronald Calabrese as the Senior Editor. The following individual involved in review of your submission has agreed to reveal their identity: Douglas Portman (Reviewer #3).

The reviewers have discussed the reviews with one another and the Reviewing Editor has drafted this decision to help you prepare a revised submission.

Summary:

All reviewers were supportive of this work, noting that the discovery of a sex-specific glia-to-neuron transdifferentiation event, and the correlation of this neuron with behavioral phenotypes, is of great interest. However, since the existence of this cell was previously suggested by Sulston, and both transdifferentiation and the role of glia as neuronal precursors have been reported previously, all reviewers felt (and I concur) that this work requires additional mechanistic insights at this point.

Essential revisions:

1) Strengthen the evidence for transdifferentiation. As currently presented, it is not completely clear whether this is direct transdifferentiation or whether a cell division event occurs. A few experimental avenues were suggested. These include:

a) Examining whether you detect an apoptotic cell in the vicinity of PHso1 during its transdifferentiation. If not, this might help bolster the direct transdifferentiation model, as opposed to division of PHso1 followed by rapid death of one of the daughters. Along these lines, also consider looking at a canonical cell death mutant.

b) Provide time lapse video of the differentiation event so that the cell dynamics can be easily followed. A reviewer felt that the images in Figure 2 were not very clear and suggested using a membrane-tagged reporter for easier visualization.

c) A number of genes have previously been shown to regulate the Y-to-PDA transdifferentiation (Jarriault and others). Determining whether these genes or a subset also play a role in the PHso1 transdifferentiation process would help to establish whether the processes are shared or distinct and begin to outline the underlying mechanisms.

2) Clarify and strengthen the behavioral analysis. The correlation of the PHD neuron with the 'Molina maneuver' is a particular strength of this work.

a) You nicely propose that the PHDs couple arched tail posture to reverse locomotion. However, one would also expect a similar requirement at the beginning of contact-response behavior, where linking an arched tail posture to reverse locomotion is also important. Please determine whether response behavior is less smooth, even if it ultimately occurs just as frequently as in wildtype.

b) In Video 4, it seems that, after moving away from the vulva, the male repeatedly attempts intromission "ectopically", near the tail. It's hard to tell from the video whether spicule prodding is happening at this time, but if it is, could it be that PHD is not required for the Molina manoeuvre itself, but rather to prevent inappropriate spicule prodding at locations away from the vulva? Please look for pausing/prodding at locations other than the vulva during scanning behavior. If this is observed, it could be that the PHDs instead inhibit Lov (location-of-vulva) behavior.

c) Videos 3, 4, S3: it would be helpful to indicate in the video itself exactly when the "Molina manoeuvre" takes place and, in Video 4, when the defect occurs. Also, it should be noted somewhere that these videos are sped up 2x from real time.

d) While the ablation data are nice, there are some issues regarding how you determined that the cells were ablated, and the absence of an orthogonal method of ablation for confirmation. Reviewers suggested a couple of possibilities for experiments: you have a PHD-specific *oig-8* promoter version. You could try to feminize driving TRA-2(IC) under this promoter and checking behavior although it is possible that this promoter comes on too late to feminize. Alternatively, you could silence/ablate PHD using this promoter to confirm the behavioral phenotype.

---

## [Author Response]

Summary:All reviewers were supportive of this work, noting that the discovery of a sex-specific glia-to-neuron transdifferentiation event, and the correlation of this neuron with behavioral phenotypes, is of great interest. However, since the existence of this cell was previously suggested by Sulston, and both transdifferentiation and the role of glia as neuronal precursors have been reported previously, all reviewers felt (and I concur) that this work requires additional mechanistic insights at this point.

We have now performed a number of additional experiments – described in more detail below – that provide several additional cellular and mechanistic insights. In summary we find that, at the molecular level, the “plasticity cassette” described by Sophie Jarriault’s lab is not required for either PHso1-to-PHD or AMso-to-AMso’+MCM. This suggests not all transdifferentiations in *C. elegans* are equivalent and that the genes involved in Y-to-PDA transdifferentiation are not the only plasticity genes. We also now provide data that demonstrate that PHso1 actually divides at low frequency and in a background dependent manner to generate PHD1/PHD2. We find this variability incredibly intriguing of course, as the only other variable lineages in *C. elegans* are the EF neurons, the DX neurons and P3.p vulval precursor cells. The EFs are the major post-synaptic target of the PHDs. We are continuing our work on both the molecular regulation and variable cell division aspects in our ongoing studies, outside the scope of this manuscript.

Although the potential existence of PHD had indeed been suggested by John Sulston, he also stated “no other neuronal characteristics were observed” aside from the presence of cilia. Here we demonstrate the acquisition of neuronal features at the morphological, ultrastructural and molecular levels as well as the function of the PHD neuron in male mating behaviour. We think that a second report of a glia-to-neuronal cell fate switch actually cements this process as a fundamental mechanistic principle of neural development (which in many ways is very similar to vertebrate adult neurogenesis). This is also only the second well-described example of a direct transdifferentiation in *C. elegans* (the other being Y-to-PDA). The number of transdifferentiation paradigms described with clear lineage relationships pre/post cell fate switch and at such cellular resolution are few and far between and so should be of interest to *eLife* readers.

Essential revisions:1) Strengthen the evidence for transdifferentiation. As currently presented, it is not completely clear whether this is direct transdifferentiation or whether a cell division event occurs. A few experimental avenues were suggested. These include:

As mentioned above, we have now performed a number of additional experiments to address this and we thank the reviewers for pushing us to look into this in much more detail. We find that ~80% of the time PHso1-to-PHD is a direct transdifferentiation. However, in ~20% of cases (variable depending on background) PHso1 actually divides to generate what we have termed PHD1 and PHD2. We note that in all our behaviour experiments we ensured following laser ablation that no PHDs remained and that the drivers we used to silence the cells would also silence PHD2. We find PHD2 indistinguishable from PHD1 in terms of gene expression.

a) Examining whether you detect an apoptotic cell in the vicinity of PHso1 during its transdifferentiation. If not, this might help bolster the direct transdifferentiation model, as opposed to division of PHso1 followed by rapid death of one of the daughters. Along these lines, also consider looking at a canonical cell death mutant.

John Sulston originally described all the apoptotic events in the male tail and in corroboration of his data we do not observe any apoptotic cell corpses in the vicinity of PHso1. However, given that the male tail is undergoing quite dramatic morphogenetic changes at the same time as PHso1-to-PHD transdifferentiation, we have also analysed *ced-4(n1164)* mutants. We find no evidence for a cell death arising from PHso1 (Figure 3—figure supplement 1). To our surprise however, at a low frequency we did observe the presence of one or two extra cells in the tail (maximum one extra cell per side) co-expressing *lin-48^prom^::tdTomato* and *rab-3^prom^::yfp* in both mutant and control animals and have now investigated the origin of these cells in more detail (see below).

b) Provide time lapse video of the differentiation event so that the cell dynamics can be easily followed. A reviewer felt that the images in Figure 2 were not very clear and suggested using a membrane-tagged reporter for easier visualization.

We now provide, in an updated version of Figure 2, a time-series of an individual animal imaged every 2–4 h over the 10–12 h period of PHso1-to-PHD transdifferentiation. We think that this experiment unambiguously demonstrates that in the majority of cases PHso1-to-PHD is a direct transdifferentiation.

Following our observation of additional cells expressing *lin-48/rab-3* in the vicinity of PHD (see above), we have now characterised in more detail the battery of glial/neuronal markers at our disposal and find the presence of an additional PHD-like cell at a low and background-dependent frequency (13-24%). This data is now presented in Figure 3—figure supplement 1. We find these extra cells indistinguishable from PHD both in terms of morphology and gene expression. To determine the origin of these cells we performed Edu staining experiments, to look for dividing cells, in a *lin-48^prom^::tdTomato* background. We only detect Edu staining in *lin-48^prom^::tdTomato* labelled cells in the tail if, and only if, two cells per side are observed. We observe this at a frequency of 21% (Figure 3). In addition, we observed the division of PHso1 “live” in one of our time-lapses of single animals (Figure 3—figure supplement 2). All together this indicates that while PHso1 mainly transdifferentiates directly into PHD it can also, at low frequency, divide symmetrically to generate two neurons, which we now refer to as PHD1 and PHD2.

c) A number of genes have previously been shown to regulate the Y-to-PDA transdifferentiation (Jarriault and others). Determining whether these genes or a subset also play a role in the PHso1 transdifferentiation process would help to establish whether the processes are shared or distinct and begin to outline the underlying mechanisms.

We have now analysed loss-of-function mutants in *sem-4* and *egl-27*, as well as both RNAi and a mosaically-rescued null allele of *sox-2*. These genes are all core members of the Y-to-PDA “plasticity cassette” which has been shown by the Jarriault lab to be required for the initiation of Y-to-PDA transdifferentiation, specifically in the erasure of Y identity. We find that these factors are largely dispensable for PHso1-to-PHD transdifferentiation. This data is presented in Figure 5, a new figure. Being somewhat surprised (but rather excited) by these results we extended our analysis to AMso-to-AMso’+MCM transdifferentiation and again find these factors are dispensable. All together our results demonstrate that AMso-to-AMso’+MCM indirect transdifferentiation and PHso1-to-PHD direct transdifferentiation are likely governed by distinct mechanisms. This suggests that the “plasticity cassette” are not the only plasticity genes in *C. elegans* and we are looking forward to discover the molecular mechanisms of glia-to-neuron cell fate switches in our future work.

2) Clarify and strengthen the behavioral analysis. The correlation of the PHD neuron with the 'Molina maneuver' is a particular strength of this work.a) You nicely propose that the PHDs couple arched tail posture to reverse locomotion. However, one would also expect a similar requirement at the beginning of contact-response behavior, where linking an arched tail posture to reverse locomotion is also important. Please determine whether response behavior is less smooth, even if it ultimately occurs just as frequently as in wildtype.

We have re-analysed the videos from the ablation experiments scoring “hesitation” as a more sensitive measurement of the quality of the Response step. As described in Materials and methods, we have termed “hesitation” a switch in direction between forward and backward locomotion from the time the male establishes contact with the mate to the first turn (or to location of vulva if this occurs without the need of a turn). Although the proportion of PHD-ablated males that hesitate during response is slightly higher than mock males, the difference is not statistically significant. One possible reason for a less pronounced discontinuity in movement during response compared to during scanning may be that we are measuring locomotion during a shorter period of time during response and so the full defects cannot reveal themselves. We have added this data to Figure 7B

b) In Video 4, it seems that, after moving away from the vulva, the male repeatedly attempts intromission "ectopically", near the tail. It's hard to tell from the video whether spicule prodding is happening at this time, but if it is, could it be that PHD is not required for the Molina manoeuvre itself, but rather to prevent inappropriate spicule prodding at locations away from the vulva? Please look for pausing/prodding at locations other than the vulva during scanning behavior. If this is observed, it could be that the PHDs instead inhibit Lov (location-of-vulva) behavior.

We have re-analysed the videos of the ablations experiments and scored ectopic prodding during Molina manoeuvres (Figure 7—figure supplement 2A and B) and during scans (Figure 7—figure supplement 2C and D). We only observed a small but statistically significant increase in ectopic prodding during Molina manoeuvres in PHD-ablated males compared to mocks (Figure 7—figure supplement 2A). However, ectopic prodding occurred exclusively during discontinuous manoeuvres and the proportion of discontinuous Molina manoeuvres with ectopic prodding was similar in mock and PHD-ablated males (Figure 7—figure supplement 2B). Therefore, we conclude that the main defect resulting from the ablation of PHD is discontinuity in locomotion during manoeuvres and that ectopic prodding may be a secondary consequence of pausing and interrupting locomotion. Because PHD-ablated males perform many more discontinuous manoeuvres than mocks, ectopic prodding also increases.

We have also changed the graph showing the percentage of discontinuous scans in Figure 7C and added categories so that the plot matches the one for Molina manoeuvres in Figure 7F.

c) Videos 3, 4, S3: it would be helpful to indicate in the video itself exactly when the "Molina manoeuvre" takes place and, in Video 4, when the defect occurs. Also, it should be noted somewhere that these videos are sped up 2x from real time.

We have added labels in those videos indicating when Molina manoeuvres and discontinuity defects occur. We have also indicated that the videos are sped up 2x in the video legend. We have also renamed the videos so that they are all main and not supplemental videos. Current Videos 1 to 7 were respectively, video 1, 2, S1, S2, 3, S3 and 4

d) While the ablation data are nice, there are some issues regarding how you determined that the cells were ablated, and the absence of an orthogonal method of ablation for confirmation. Reviewers suggested a couple of possibilities for experiments: you have a PHD-specific oig-8 promoter version. You could try to feminize driving TRA-2(IC) under this promoter and checking behavior although it is possible that this promoter comes on too late to feminize. Alternatively, you could silence/ablate PHD using this promoter to confirm the behavioral phenotype.

We respectfully disagree with the reviewer’s view that there were issues regarding how we determine the PHD cells are ablated. We believe laser ablations are the cleanest and most specific way to remove neurons in an otherwise intact animal and we have extensive expertise in larval ablations. Any other type of manipulation that relies on a genetic driver has the caveat of lack of specificity without a single-cell-specific driver. The most specific promoter at our disposal, the *oig-8* promoter, is not uniquely expressed in PHD but also in two classes of sensory neurons in the head. To further demonstrate the specificity of our laser ablations and the method by which we ensure PHD cell death, we have included a new ablation control experiment in which we specifically ablate PHDR, leaving PHDL intact (the cells are ~5µm apart). These control experiments are presented in Figure 7—figure supplement 1.

We have additionally pursued a genetic approach as requested by the reviewers. We initially tried driving a reconstituted *ced-3* caspase in the PHD neurons but we either failed to get transgenic lines at mid-range concentrations (15 ng/µl) or failed to efficiently kill all PHD neurons in a significant proportion of animals at low concentrations (5 ng/µl). We then generated an inducible silencing transgene driving an histamine-gated chloride channel in PHD neurons*, oig-8::HisCl1::sl2::rfp*. We could confirm expression of this transgene in PHD neurons by red fluorescence. This approach recapitulated the discontinuity in Molina manoeuvres that we had previously observed in the laser ablation experiments. We have included this data in Figure 7F. However, we would like to note that we did not observe a statistically significant difference in the quality of scans (discontinuous versus discontinuous) in PHD-silenced males compared to controls (data not shown). We don’t know the reasons for a lack of a scan phenotype in the silencing experiments but having extra oig-8 expressing neurons silenced in the head may compensate for the absence of PHD activity. Alternatively, removing the neuron completely through ablation may impact the circuit more severely than just silencing it.